# Antibody-mediated spike activation promotes cell-cell transmission of SARS-CoV-2

Shi Yu[1☯¤], Xu Zheng[1☯], Yanqiu Zhou[2☯], Yuhui Gao[1], Bingjie Zhou[1], Yapei Zhao[1], Tingting Li[3], Yunyi Li[2], Jiabin Mou[2], Xiaoxian Cui[2], Yuying Yang[2], Dianfan Li[3], Min Chen[2]*, Dimitri Lavillette[1,4,5]*, Guangxun Meng[1,4,6]*

1 The Center for Microbes, Development and Health, CAS Key Laboratory of Molecular Virology & Immunology, Shanghai Institute of Immunity and Infection, University of Chinese Academy of Sciences, Shanghai, China, 2 Shanghai Municipal Center for Disease Control and Prevention, Shanghai, China, 3 State Key Laboratory of Molecular Biology, State Key Laboratory of Cell Biology, CAS Center for Excellence in Molecular Cell Science, Shanghai Institute of Biochemistry and Cell Biology, Chinese Academy of Sciences (CAS), Shanghai, China, 4 Pasteurien College, Soochow University, Suzhou, Jiangsu, China, 5 Applied Molecular Virology Laboratory, Discovery Biology Department, Institut Pasteur Korea, Gyeonggi-do, South Korea, 6 Nanjing Advanced Academy of Life and Health, Nanjing, Jiangsu, China

☯ These authors contributed equally to this work.
¤ Current address: Guangzhou National Laboratory, Department of Basic Research, Guangzhou International Bio-Island, Guangzhou, China.
* chenmin@scdc.sh.cn (MC); dimitri.lavillette@ip-korea.org (DL); gxmeng@ips.ac.cn (GM)

**Data Availability Statement:** All relevant data are within the manuscript and its Supporting Information files.

**Funding:** This work is supported by grants from Prevention and Control of COVID-19 Program

## Abstract

The COVID pandemic fueled by emerging SARS-CoV-2 new variants of concern remains a major global health concern, and the constantly emerging mutations present challenges to current therapeutics. The spike glycoprotein is not only essential for the initial viral entry, but is also responsible for the transmission of SARS-CoV-2 components via syncytia formation. Spike-mediated cell-cell transmission is strongly resistant to extracellular therapeutic and convalescent antibodies via an unknown mechanism. Here, we describe the antibody-mediated spike activation and syncytia formation on cells displaying the viral spike. We found that soluble antibodies against receptor binding motif (RBM) are capable of inducing the proteolytic processing of spike at both the S1/S2 and S2' cleavage sites, hence triggering ACE2-independent cell-cell fusion. Mechanistically, antibody-induced cell-cell fusion requires the shedding of S1 and exposure of the fusion peptide at the cell surface. By inhibiting S1/S2 proteolysis, we demonstrated that cell-cell fusion mediated by spike can be re-sensitized towards antibody neutralization *in vitro*. Lastly, we showed that cytopathic effect mediated by authentic SARS-CoV-2 infection remain unaffected by the addition of extracellular neutralization antibodies. Hence, these results unveil a novel mode of antibody evasion and provide insights for antibody selection and drug design strategies targeting the SARS-CoV-2 infected cells.

## Author summary

SARS-CoV-2 has been found to mediate breakthrough infections in vaccinated individuals and re-infections of convalescent patients, but the molecular mechanism of its immune

(2020T130119ZX) of Postdoctoral Science Foundation of China and International Postdoctoral Exchange Fellowship (251371) (SY); Ministry of Science and Technology of China (2022YFE0114700) and the CAS president's international fellowship initiative (2020VBA0023) of the Chinese Academy of Sciences (DL); Strategic Priority Research Program (XDB29030303), Shanghai Municipal Science and Technology Major Project (2019SHZDZX02, 20431900402) and Research Leader Program (20XD1403900), National Key R&D Program of China (2022YFC2303200, 2022YFC2304700, 2020YFC0845900) (GM); Natural Science Foundation of China (82151215, DLi; 31870153, DL;). The funders had no role in study design, data collection and analysis, decision to publish, or preparation of the manuscript.

**Competing interests:** The authors have declared that no competing interests exist.

escaping strategies remains elusive. Unlike virus-to-cell transmission, spike promotes SARS-CoV-2 cell-to-cell transmission, which is strongly resistant to extracellular antibodies and patient plasma. Here we show that receptor binding-motif antibodies mediate the ACE2-independent activation of spike at the cell plasma membrane. This mode of spike activation is enabled by the protease-mediated S1/S2 cleavage event and can be genetically and pharmacologically prevented. Through targeting the S1/S2 site, antibody neutralization against spike-mediated cell-cell fusion can be restored in various SARS-CoV-2 variants. Hence, these data highlight a role for S1/S2-cleaved spike and inform therapeutic strategies to restore antibody neutralization against cell-cell transmission of SARS-CoV-2.

## Introduction

Severe acute respiratory syndrome coronavirus 2 (SARS-CoV-2) is the causative agent of coronavirus disease (COVID), and the emerging new variants are extending the global spread and threatening the future vaccine efficacy. The SARS-CoV-2 spike (S) glycoprotein is a class I fusion protein decorated on the viral envelope and is a key determinant of viral entry [1]. Targeting spike and its function is of great interest for many therapeutic approaches. However, SARS-CoV-2 is known to mediate breakthrough infections in the vaccinated individuals, as well as re-infections of convalescent patients recovered from prior infection, but the molecular mechanism of its immune escaping strategies remain elusive.

The SARS-CoV-2 spike protein contains two fragments: the amino-terminus S1 subunit contains a receptor binding domain (RBD) [2–5], which recognizes host receptor angiotensin-converting enzyme 2 (ACE2) at the plasma membrane for its initial docking; while subsequent conformational rearrangement of S2 subunit catalyzes the fusion of viral and cell membranes [6,7], ultimately enables the release of viral RNA genome and downstream replication within the infected cells [8]. Compared to the original SARS-CoV, the SARS-CoV-2 spike carries a polybasic cleavage site at the S1/S2 junction and is proteolytically processed by host furin at the arginine 685 (R685) during its protein synthesis [9,10]. Although SARS-CoV-2 spike is cleaved at S1/S2 during biosynthesis, S1 remains non-covalently attached to the S2 [1], which requires further receptor priming and subsequent cleavage at the S2' site to mediate downstream membrane fusion [11,12].

Functionally, polybasic residues at the spike S1/S2 junction strongly promote cell-cell fusion among ACE2-expressing cells, a feature termed as syncytia formation [9,13,14]. Syncytia formation among pneumocytes has been described as a clinical hallmark in the lung pathogenesis of SARS-CoV-2 infection [15–18]. Compared to the ancestral Wuhan strain, residues adjacent to the furin cleavage site acquired additional mutations in current variants of concern (VOCs), including the P681H mutation reported from the SARS-CoV-2 Alpha and Omicron subvariants, as well as the P681R mutation in Delta and some lineage A variants [19]. These polybasic residues render the S1/S2 cleavage site highly susceptible to additional proteases and accelerate the syncytium formation. The cleaved S1 subunit C-terminus also facilitates the binding of attachment receptors, such as neuropilin-1 (NRP1), to further promote the infection of lung epithelial cells in human and other animal models [9,10,20–24]. It is currently unclear whether additional cellular processes could trigger the proteolytic processing at S1/S2, and the functional role of S1/S2 cleavage site in terms of infectivity and transmissibility remains elusive.

Critically ill and hospitalized COVID patients display high levels of anti-spike immunoglobulin G (IgG) antibodies compared to mild, non-hospitalized control and convalescent

patients [25,26]. More specifically, these elevated IgGs recognize the RBD of spike S1 subunit, primarily at the receptor binding motif (RBM) and display potent ACE2-blocking properties. Moreover, *in vitro* neutralization assays demonstrated that these RBM antibodies are highly effective in neutralizing SARS-CoV-2 viral particles, many have been structurally resolved at atomic resolution. Antibody molecular mimicry has been proposed as a mechanism for their neutralization actions against the viral spike [27–29]. Although RBM antibodies are effective at neutralizing SARS-CoV-2 viral particles, spike-mediated cell-cell transmission of the virus remains resistant to neutralization and convalescent antibodies [30–33]. It is also unclear whether excess anti-spike IgGs may play an adverse or unfavorable role on infected cells in terms of driving the host pathology.

Here, we examined various types of anti-RBD neutralization antibodies on cells expressing ancestral SARS-CoV-2 WT and VOC spike proteins. Combining qualitative and quantitative cell-cell fusion models, we found that RBM-specific antibodies triggered cell-cell fusion in the absence of ACE2, despite these antibodies carry strong neutralization capacity against viral particles. Moreover, we showed that antibody-induced cell-cell fusion requires the cleavage at the S1/S2 site. Mechanistically, the cleaved S1/S2 site triggers the shedding of soluble S1 upon RBM antibody binding, enabling the activation of S2' site at the plasma membrane. We also demonstrated that antibody-induced cell-cell fusion is absent in furin cleavage site-deficient spike, and can be genetically abolished by furin knock-down and pharmacological inhibition. Since cellular spike utilizes the antibody molecular mimicry to evade RBM targeting neutralization antibodies, our work revealed that targeting the S1 shedding event could re-sensitize antibody neutralization against spike-mediated cell-cell fusion.

## Results

### Receptor binding motif (RBM) antibodies drive cell-cell fusion in the absence of ACE2

To examine the effect of antibodies on spike-expressing cells, we selected soluble SARS-CoV-2 neutralizing antibodies that developed against the RBD to block its receptor usage [34–39]. One of the first human SARS-CoV-2 monoclonal antibody isolated from convalescent patient —CB6 (Class I, also known as LY-CoV016 or Etesevimab, $K_D$ ~2.49 nM), recognizes epitopes of the receptor binding motif (RBM) and competitively blocks spike and ACE2 interaction [34]. For comparison, we selected FD20 (Class IV, $K_D$ ~5.60 nM), a human monoclonal antibody that recognizes cryptic epitopes distal to the RBM [40]. We also used the FDA-approved REGN10933 (Class I, casirivimab, $K_D$ ~3.37 nM) and REGN10987 (Class II&III, imdevimab, $K_D$ ~45.2 nM) that target RBM and non-RBM interfaces of the ancestral strain, respectively [41,42]. We initially tested this panel of neutralization IgGs and investigated their potential capability in triggering the spike-mediated cell-cell fusion in ACE2-deficient HEK293T control cells (**Fig 1A**). We quantified the bioluminescence readings in cell lysates after adding 12.5 nM of these human antibodies on HEK293T cells co-expressing ancestral WT spike and Cre (donor cells), mixed with HEK293T cells expressing *Stop-Luc* (acceptor cells) for 16 hours. Intriguingly, we detected robust increase in bioluminescence signal reflecting cell-cell fusion from the Class I CB6- and REGN10933-treated cell lysates when compared to no antibody control samples, or Class IV FD20- and Class III REGN10987-treated cell lysates (**Fig 1B**). Antibody-induced cell-cell fusion did not result in cell apoptosis, since no poly ADP-ribose polymerase (PARP) cleavage was detected (**S1A Fig**).

To validate whether antibody-induced cell-cell fusion occurs on human lung-associated epithelial cells, we prepared a lentiviral transfer vector co-expressing SARS-CoV-2 full-length wildtype (WT) spike in the WT A549 cells (**Fig 1C**). 48 hours after lentiviral transduction, we

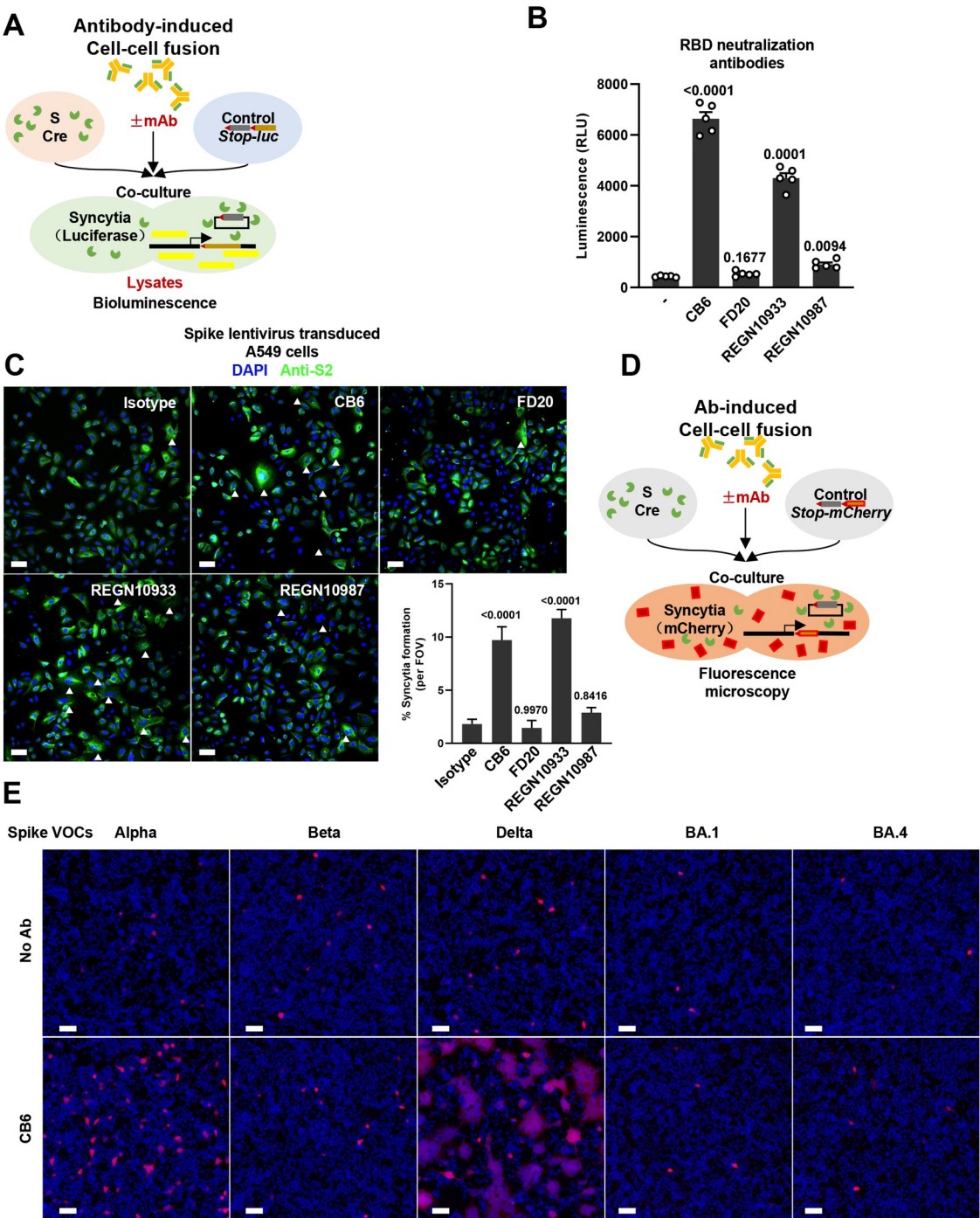

**Fig 1. Receptor binding motif (RBM) antibodies drive ACE2-independent cell-cell fusion** (**A**) Schematics of the antibody-induced cell-cell fusion model used to quantify spike-mediated syncytium formation. Cells co-expressing SARS-CoV-2 spike and Cre, were co-cultured with *Stop-luc* expressing HEK293T cells for 16 hours, before cell lysates were collected for bioluminescence detection; (**B**) Luciferase activity (RLU) measured from HEK293T cell lysates collected from the neutralization antibody stimulation described in (A) for 16 hours. CB6, FD20, REGN10933 and REGN10987 were diluted in PBS and used at 12.5 nM. Data are representative of five individual repeats. Data are displayed as individual points with mean ± standard error of the mean (SEM); (**C**) Representative fluorescent image captured at 488 nm from lentivirus-transduced A549 cells expressing SARS-CoV-2 spike, stimulated with 12.5 nM Isotype, CB6, FD20, REGN10933 and REGN10987 for 16 hours. Anti-S2 was stained with Alexa fluor 488, syncytia are indicated with white arrows and syncytia formation per field of view (FOV) from individual repeats were summarized, scale bars are representative of 50 μm. Images were representative of at least four individual repeats; (**D**) Schematics of the antibody-induced cell-cell fusion model for qualification of spike-induced syncytia formation. Cells co-expressing spike and Cre, were

cocultured with *Stop-mCherry* expressing HEK293T cells for 16 hours, cell nuclei were counterstained with 100 ng/mL Hoescht33342 and fluorescent images were then captured; (**E**) Representative fluorescent images for mCherry reporter and Hoechst33342 captured at 594 and 405 nm from HEK293T cells expressing Alpha, Beta, Delta, Omicron BA.1 and BA.4 spike VOCs, stimulated without or with 12.5 nM CB6 antibody, scale bars are representative of 50 μm. Images are representative of three individual experiments.

stimulated spike lentivirus transduced A549 cells with the panel of RBD antibodies and examined the effect of antibody-treatment on syncytia formation using immunofluorescence. In line with our observation on HEK293T cells, when compared to control, addition of CB6 and REGN10933 also significantly induced syncytia formation in the spike-expressing A549 cells (**Fig 1C**), with clustered, multinucleated cells displaying enlarged and expanded cytoplasm.

Unlike ACE2 receptor that contains a transmembrane anchor, CB6 is a soluble human immunoglobulin G (IgG1) [34]. When antibodies are added to the spike-expressing cells, CB6 may directly connect adjacent spike-expressing cells for membrane fusion (Crosslinking syncytia); otherwise, CB6 may trigger the activation of spike to fuse with random adjacent control cells (Alternative syncytia). We tested these two scenarios by co-transfecting fluorescent reporter ZsGreen and spike in donor, or mCherry in the acceptor cells; quantification of the CB6 crosslinked green syncytia and alternatively-activated yellow syncytia can be directly visualized after antibody treatments (**S1B Fig**). We found that although CB6-induced noticeable crosslinking between ZsGreen expressing cells, alternatively-activated syncytia were also formed between spike and control mCherry cells (**S1B Fig**); no mCherry syncytia were observed. These results demonstrated that CB6-induced spike activation causes cell-cell fusion between spike-expressing cells, but also non-specifically among adjacent or bystander cells.

To further validate and visualize the specificity of CB6 antibody induced syncytia formation among bystander cells, we modified the bioluminescence system described in Fig 1A and replaced the firefly luciferase encoding gene with a mCherry fluorescent reporter to qualitatively determine spike activation by CB6 antibody (**Fig 1D**). Similar to the WT spike, CB6-induced cell-cell fusion in Alpha and Delta spike VOCs, but failed to induce syncytia formation in cells expressing Beta, Omicron BA.1 and BA.4 spike (**Fig 1E**). Notably, CB6 displayed loss of binding to Beta spike carrying the K417N substitution as we previously reported [40], or multiple substitution changes in spike Omicron BA.1 and BA.4 subvariants (**S1C Fig**). Hence, syncytia formation induced by neutralization antibody likely requires specific recognition and binding of the spike RBM and occurs on cells expressing SARS-CoV-2 spike.

To show antibody-induced syncytia formation and viral particle neutralization are two independent processes, we utilized a murine leukemia virus (MLV) packaging system to simultaneously visualize and assess antibody neutralization of the packaged pseudotyped particles (PPs) on spike-expressing cells. We performed the packaging of WT and Delta spike PPs where CB6 were added in the culture supernatants 6 hours after co-transfecting the MLV packaging components. Fluorescent images captured 24 hours after CB6 treatment showed robust syncytia formation among PPs packaging cells (**S1D Fig**); while infectivity of the supernatants containing CB6-treated PPs on HEK293T-ACE2 cells were completely neutralized when compared to the control WT and Delta spikes pseudotype particles (**S1E Fig**). These data suggested that, spike expressed on the plasma membrane during viral assembly could functionally engage RBM neutralizing antibodies to promote cell-cell fusion.

## Antibody-induced cell-cell fusion requires spike proteolysis at the S1/S2 junction

Spike expressed on cells exist as stable homotrimers, until binding to the functional receptor ACE2 for its proteolytic activation and subsequent membrane fusion [12]. Since RBM

antibodies induced cell-cell fusion among control HEK293T cells in the absence of ACE2, we tested whether RBM antibodies could also induce the proteolytic activation of spike due to receptor molecular mimicry. Using the same panel of antibodies described in Fig 1B, we collected both cell lysates and supernatants after stimulation to examine possible cleaved spike protein products using immunoblots. As expected, we detected potent release of cleaved S1 subunits into the cell supernatants in samples treated with CB6 and REGN10933, but to a much lesser extent in the FD20- and REGN10987-treated cells (**Fig 2A, top**). More importantly, soluble CB6 and REGN10933 induced robust S2' cleavage, a proteolytic product that is associated with membrane fusion [12], but not by FD20 or REGN10987 at the same concentration (**Fig 2A, bottom**). CB6- and REGN10933-induced cleavage of S2' products were highly resistant to limited proteolysis in the presence of 10 μg/mL proteinase K, suggesting that binding of CB6 and REGN10933 on spike expressing cells readily converted spike into the postfusion conformations reminiscent of the rigid 6-helix bundles [12] (**Fig 2B**). These data suggested that soluble RBM antibodies mimic the ACE2 receptor binding and lead to spike S1 shedding, as well as S2' cleavage.

Using the antibody-induced cell-cell fusion model, we functionally examined the dose response of CB6 and REGN10933 on cell-cell fusion and spike proteolysis on HEK293T cells. CB6-(**Fig 2C**) and REGN10933-induced (**S2A Fig**) cell-cell fusion, S1 shedding and S2' cleavage were concentration dependent, but not by a human control isotype IgG at the highest dose of 12.5 nM (**Figs 2C and S2A**). To validate antibody-induced spike cleavage also occurs on lung-associated epithelial cells, we found that CB6 and REGN10933 also induced the shedding of S1 subunit into the culture supernatant of spike-transduced A549 cells (**Fig 2D**). To test whether CB6-induced cell-cell fusion requires the functional refolding of S2 subunit, we used a HR1 peptide inhibitor EK1C4 [43] (Gift from Prof Shibo Jiang). 2.5 μM EK1C4 potently blocked CB6-induced cell-cell fusion on cells expressing WT spike and had no effect on S1 shedding and S2' cleavage (**S2B Fig**). These data suggested that CB6-induced spike proteolysis at S1/S2 and S2' are upstream of membrane fusion.

To highlight the requirement of specific RBM recognition, we generated a N417K reversion substitution on the spike Beta VOC. Similar to Fig 1E, addition of 12.5 nM CB6 did not induce cell-cell fusion of the Beta spike (N417), but robustly induced cell-cell fusion from the Beta spike K417 mutant, demonstrated by the significant increase in bioluminescence (**Fig 2E, top**), as well as the expression of mCherry reporter (**S2C Fig**). The single reversion substitution was sufficient to restore CB6-induced spike proteolysis, since S1 shedding in the culture supernatants and the S2' cleavage in cell lysates were robustly detected (**Fig 2E, bottom**). Therefore, these results demonstrated that RBM antibody induced cell-cell fusion is strongly associated with the spike proteolysis at S1/S2 and S2' cleavage sites, and require specific binding onto the spike RBM.

To investigate the involvement of ACE2 expressed on HEK293T and A549 cells, we utilized two strategies to rule-out the roles of possible endogenous functional receptor. First, we tested the ability of CB6 to induce spike-mediated cell-cell fusion in the presence of a mouse monoclonal anti-ACE2 blocking antibody. Anti-ACE2 completely inhibited SARS-CoV-2 MLV PPs infection of the HEK293T-ACE2 cells at 10 μg/mL (**S2D Fig**), but had no enhancing, nor inhibiting effect on CB6-induced cell-cell fusion on spike-expressing HEK293T control cells (**S2E Fig, top**). Anti-ACE2 blocking antibody also had no effect on shedding of S1, as well as the S2' cleavage induced by the CB6 antibody (**S2E Fig, bottom**). Secondly, we performed knockout of endogenous ACE2 using CRISPR-Cas9 technology in HEK293T. Similar to the ACE2 blocking antibody, amount of CB6-induced cell-cell fusion in *sgControl* and *sgACE2* HEK293T were consistent with WT cells used (**S2F Fig**). Moreover, in A549 cells transduced with spike, no effect on CB6-induced S1 shedding when anti-ACE2 blocking antibody was

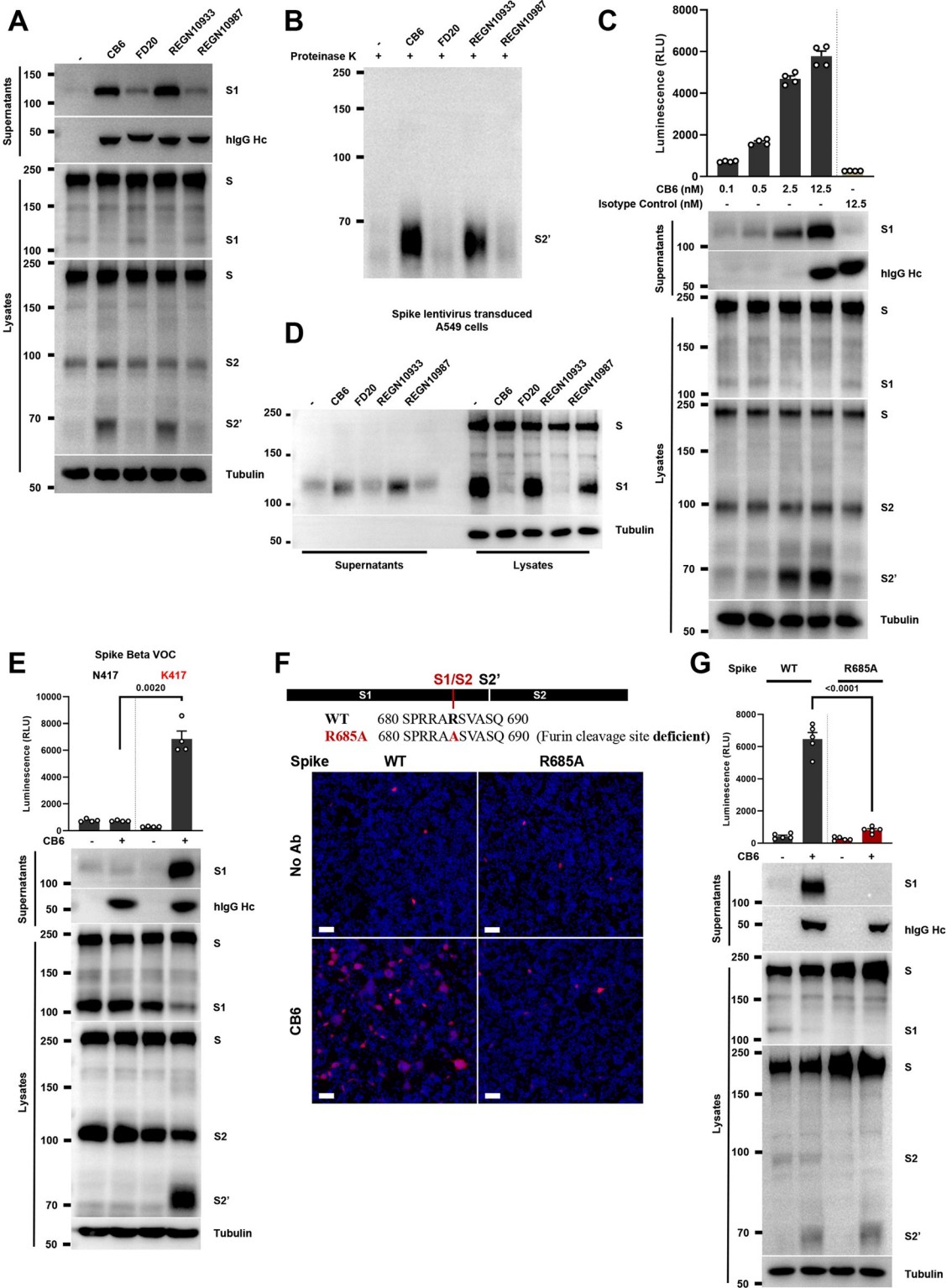

**Fig 2. Antibody-induced cell-cell fusion requires spike proteolysis at the S1/S2 junction.** (**A**) Immunoblots showing shedded S1 subunits and human IgG heavy chain (IgG Hc), full-length spike, S1, S2 and cleaved S2' collected from supernatant and cell lysate fractions of antibody-treated HEK293T cells expressing WT full-length spike for 16 hours. CB6, FD20, REGN10933 and REGN10987 were diluted in PBS and used at 12.5 nM, blots are representative of at least three independent experiments; (**B**) Immunoblots showing proteinase K-resistant S2' cleavage product, obtained from cell lysates described in (A) and were then treated in the absence

or presence of 10 μg/mL proteinase K for 30 min at 37°C, blots are representative of two individual repeats; (**C**) Luciferase activity (RLU) measured from cell-cell fusion assay and immunoblots showing shedded S1 subunits, IgG Hc, full-length spike, S1, S2 and cleaved S2' collected from supernatant and cell lysate fractions of CB6-stimulated HEK293T cells. CB6 doses used were 0.1, 0.5, 2.5 and 12.5 nM respectively, human IgG isotype was used at 12.5 nM for the negative control. Data and blots are representative of four individual repeats; (**D**) Immunoblots showing shedded S1 subunits, full-length spike and S1 collected from supernatant and cell lysate fractions of lentivirus-transduced A549 cells expressing WT full-length spike, stimulated without or with 12.5 nM CB6 antibody. Blots are representative of three individual repeats; (**E**) Luciferase activity (RLU) measured from cell-cell fusion assay and immunoblots showing shedded S1 subunits, IgG Hc, full-length spike, S1, S2 and cleaved S2' collected from supernatant and cell lysate fractions of CB6-stimulated HEK293T cells expressing Beta spike VOCs carrying N417 or K417 revert mutation, blots and data are representative of four individual repeats; (**F**) Schematics of the R685A (furin-cleavage site deficient) spike mutant, and representative fluorescent images captured at 594 nm and 405 nm from HEK293T cells expressing WT and R685A spike mutant, stimulated without or with 12.5 nM CB6; scale bars are representative of 50 μm, images were representative of two individual repeats; (**G**) Luciferase activity (RLU) measured from antibody-induced cell-cell fusion assay, where co-cultured HEK293T cells expressing WT or R685A spike mutant were stimulated without or with 12.5 nM CB6 for 16 hours (Top). Supernatants and cell lysates were used for immunoblots of shedded S1 subunits, IgG Hc, full-length spike, S1, S2 and cleaved S2' (Bottom). Blots are representative of three independent experiments.

used (**S2G Fig**), or when *sgACE2* knockout was performed (**S2H Fig**). These data confirmed that effect of CB6-induced cell-cell fusion among spike expressing cells is ACE2 independent.

We previously generated a R685A spike mutant deficient in the furin cleavage site to investigate SARS-CoV-2 spike induced cell-cell fusion without the S1/S2 cleavage [12]. Intriguingly, the spike with furin cleavage site R685A mutation did not respond to 12.5 nM CB6 stimulation, while cells expressing WT spike readily formed syncytia and expressed the mCherry fluorescent reporter (**Fig 2F**). Verification with the cell-cell fusion experiment showed significantly reduced bioluminescence detected from CB6-treated R685A spike cell lysates (**Fig 2G, top**). As expected, shedded S1 collected from CB6-treated cell supernatants was abolished in cells expressing the R685A mutant spike (**Fig 2G, bottom**), confirming that the furin-mediated proteolytic cleavage at P1 position of the S1/S2 junction was required for the shedding of S1 upon CB6 antibody treatment. Intriguingly, CB6 induced S2' cleavage in WT spike, but also to a similar extend in the R685A spike mutant (**Fig 2G, bottom**). CB6-induced S2' cleavage in WT and R685A spike mutants were both resistant to proteinase K degradation (**S3A Fig**. These data suggested that CB6 binding to the both WT and R685A spike readily converted the spike to the post-fusion conformation, but the S1/S2 cleavage specifically and functionally enabled the cell-cell fusion induced by CB6 antibody.

## Furin-dependent priming of spike at the plasma membrane promotes antibody-mediated cell-cell fusion

Extracellularly added CB6 induced the cleavage event at the spike S1/S2 cleavage site, leading to S1 shedding into the culture supernatants. Hence, we hypothesized that in the presence of S1/S2 cleavage, antibody triggered spike S2' activation could occur at the plasma membrane; if S1/S2 cleavage does not occur, binding of CB6 could induce the internalization of spike into the endosomes, where spike S2' could be then cleaved by cathepsins [44]. Using WT and R685A spike mutants, we found that CB6 treatment strongly induced the release of S1 into the culture supernatants, along with extracellular CB6 IgG heavy chain (**Fig 3A, top**); while we detected no further cleavage of S1 and CB6 IgG in cell lysates of WT cells. In contrast, CB6 treatment of cells expressing the R685A spike mutant showed no cleavage of full-length spike in the cell lysates, nor culture supernatants; yet we detected noticeable amount of CB6 IgG heavy chain from the cell lysates (**Fig 3A, bottom**), suggested that binding of CB6 onto spike without the furin-cleaved S1/S2 site led to antibody internalization into the cells; both WT and R685A spike exhibited the cleavage of S2' similar to the Fig 2G. Moreover, when organelle acidification was inhibited by a vacuolar-ATPase inhibitor-bafilomycin A1 (BafA1) [45], we

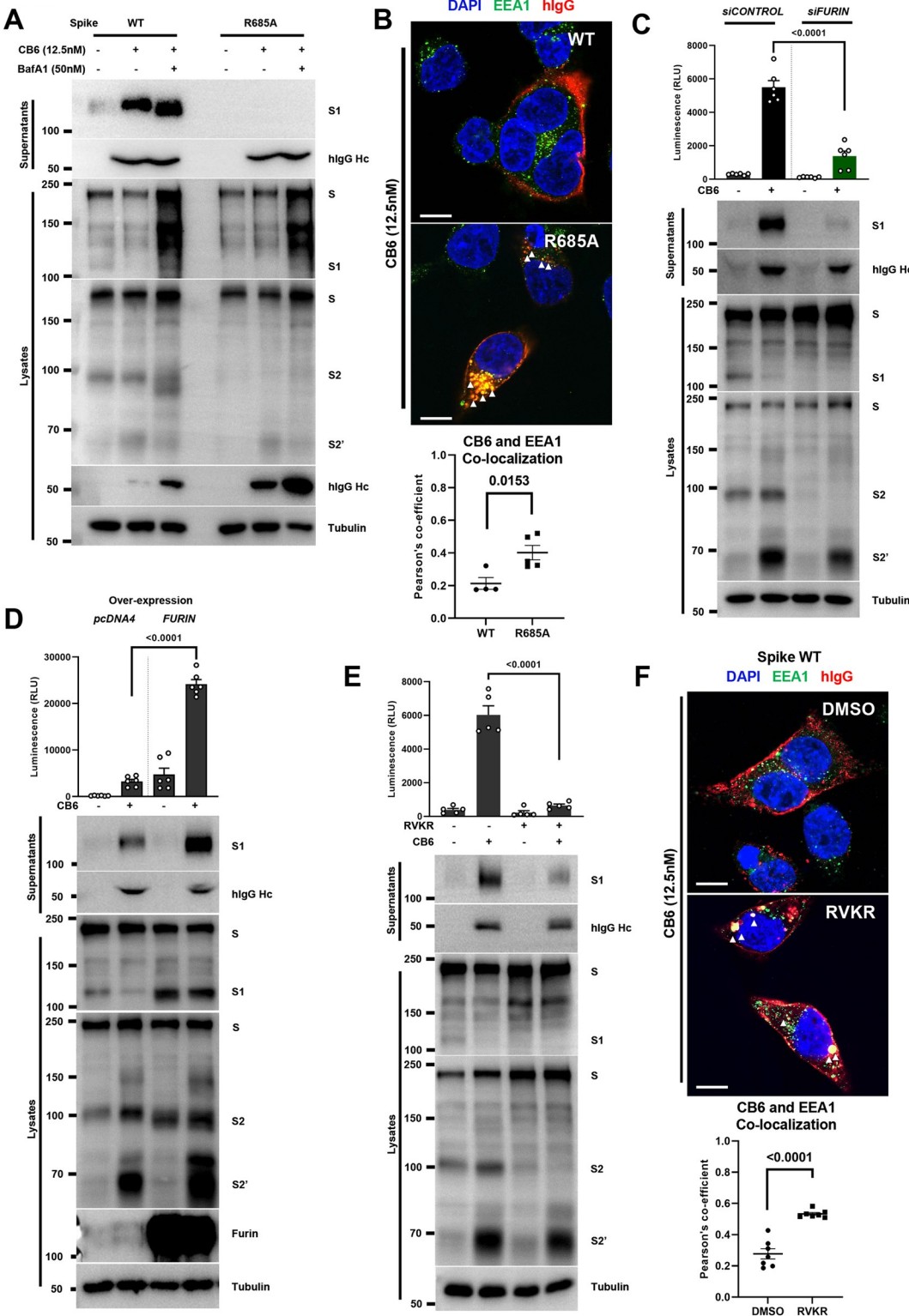

**Fig 3. Furin-dependent priming of spike at the plasma membrane promotes antibody-mediated cell-cell fusion. (A)** Immunoblots of shedded S1 subunits, IgG Hc collected from supernatants; or full-length spike, S1, S2 and cleaved S2' and IgG Hc collected from HEK293T cell lysates expressing WT or R685A spike mutant, stimulated without or with 12.5 nM CB6 antibody for 16 hours, in the absence or presence of 50 nM Bafilomycin A1 (BafA1). Blots are representative of three individual experiments; (**B**) Representative confocal images of 12.5 nM CB6 antibody-stimulated HEK293T cells expressing

WT or R685A spike mutant for 16 hours. Anti-Early endosomes antigen 1 (EEA1) and Anti-human IgG (H+L chains) were stained with Alexa fluor 488 and 555 respectively, co-localizations are indicated with white arrows and R values derived from Pearson's coefficients summarized, scale bars are representative of 10 μm. Images are representative of four individual experiments; (**C**) Luciferase activity (RLU) measured from 12.5 nM CB6-stimulated *siControl* or *siFURIN* donor HEK293T cells co-expressing Cre and WT spike, mixed with *siControl* or *siFURIN* acceptor *Stop-Luc*-expressing cells for 16 hours (top); and immunoblots showing shedded S1 subunits, hIgG Hc and full-length spike, S1, S2 and S2' collected from co-cultured cell supernatants and lysates (bottom). Data shown are representative of four independent repeats; (**D**) Luciferase activity (RLU) measured from 12.5 nM CB6-stimulated over-expressing pcDNA4 (empty vector) or FURIN donor HEK293T cells co-expressing Cre and WT spike, mixed with over-expressing pcDNA4 or FURIN acceptor *Stop-Luc*-expressing cells for 16 hours (top); and immunoblots showing shedded S1 subunits, hIgG Hc and full-length spike, S1, S2 and S2' collected from co-cultured cell supernatants and lysates (bottom). Data shown are representative of four independent repeats; (**E**) Luciferase activity (RLU) measured from HEK293T cells co-expressing WT spike and Cre, mixed with *Stop-Luc*-expressing cells, in the absence or presence of 25 μM dec-RVKR-cmk (RVKR) for 16 hours (top); immunoblots showing shedded S1 subunits, full-length S, S1, S2 and cleaved S2' collected from supernatants and lysates (bottom). Data shown are representative of five independent repeats, and the blot is representative of three repeats; (**F**) Representative confocal images of 12.5 nM CB6 antibody-stimulated HEK293T cells expressing WT spike, treated with DMSO or 25 μM RVKR for 16 hours. Anti-Early endosomes antigen 1 (EEA1) and Anti-human IgG (H+L chains) were stained with Alexa fluor 488 and 555 respectively, scale bars are representative of 10 μm, co-localizations are indicated with white arrows and R values derived from Pearson's coefficients were summarized. Data are displayed as individual points with mean ± standard error of the mean (SEM). *P* value was obtained by one-way ANOVA with Sidak's *post hoc* test and is indicated on the figure.

detected increased intracellular CB6 IgG in cells expressing WT spike, but recovered significantly more CB6 IgG in cells expressing the R685A spike mutant (**Fig 3A, bottom**).

To validate the internalization of CB6 into the cells expressing the spike furin-cleavage site mutant, we performed confocal microscopy on paraformaldehyde fixed, saponin permeabilized HEK293T cells, and examined the potential colocalization of CB6 with endosomal marker early endosome antigen 1 (EEA1). While WT spike-expressing cells displayed exclusive localization of CB6 IgGs (Red) on the syncytia plasma membrane with almost no colocalization with EEA1 (Red) observed (**Fig 3B, top**); in contrast, significant level of CB6 IgGs and EEA1 colocalization were detected in cells expressing the R685A spike mutant (**Fig 3B, bottom**). Immunostaining of human IgG further confirmed that CB6 also colocalized with LAMP1 (**S3B Fig**), a late endosome and lysosome marker, only in the spike R685A. These data suggest that CB6 induced the WT spike S1/S2 cleavage and shedding of S1 subunit at the plasma membrane, while binding CB6 to R685A spike did not trigger S1 shedding and is likely to be internalized into the cells.

Since CB6 antibody induced cell-cell fusion requires the cleavage at S1/S2 site, we next validated the genetic knock-down of host cell furin and its effect on antibody-mediated cell-cell fusion (**S3C Fig**). Compared to the *siControl*, small interference RNA knockdown of furin in both donor and acceptor cells showed significantly impaired ability in S1 shedding when 12.5 nM concentration of CB6 was added (**Fig 3C, bottom**). As a result, ACE2-independent cell-cell fusion induced by CB6 antibody was significantly inhibited in the *siFURIN* cells (**Fig 3C, top**). Conversely, when furin was overexpressed in both donor and acceptor HEK293T cells, CB6-induced S1 shedding and S2' cleavage were significantly enhanced compared to the vector control (**Fig 3D, bottom**). Enhancement of spike processing due to furin overexpression had also increased the bioluminescence from the cell-cell fusion assay (**Fig 3D, top**) and was visualized macroscopically (**S3D Fig**). Moreover, pharmacological inhibition of furin activity using 25 μM dec-RVKR-cmk (RVKR) furin inhibitor, reduced CB6-induced S1 shedding and cell-cell fusion mediated by the WT spike (**Fig 3E**). Confocal images of CB6 and EEA1 confirmed that use of RVKR restored significant level of CB6 with EEA1 colocalization in cells expressing the WT spike (**Fig 3F**). These data suggested that in the absence of S1/S2 cleavage, binding of CB6 to R685A spike may trigger the subsequent S2' cleavage by lysosomal cathepsins. These data confirmed that furin-mediated S1/S2 cleavage and S1 shedding from plasma membrane is essential for RBM antibody-driven cell-cell fusion.

Since the Delta spike VOC demonstrated strong cell-cell fusion when stimulated with CB6 (**Fig 1E**), we tested whether increased cell-cell fusion is associated with the antibody-induced s1 shedding in the spike Delta VOC. Despite numerous mutations in the spike Delta variant, CB6 still strongly immunoprecipitated the full-length Delta spike from cell lysates, but not the Omicron BA.1 spike (**S1C Fig**). As a result, CB6-induced S1 shedding and the S2' cleavage were significantly enhanced in cells expressing the Delta spike variant compared to the WT strain (**S3E Fig** bottom), as well as ACE2-independent cell-cell fusion in the Delta spike variant (**S3E Fig** top). Hence, these data revealed that polybasic residues around S1/S2 cleavage site promoted S1 shedding and antibody-induced cell-cell fusion.

## Inhibition of furin-mediated cleavage restores the neutralization efficiency of RBM antibodies

As CB6 alone was sufficient to trigger spike activation (**Figs 2 and 3**), S1 shedding event could exert a profound impact on antibody neutralization of ACE2-expressing cells in the cell-cell fusion model. In PPs neutralization assays, CB6 was serial diluted and pre-incubated with PPs prepared with WT and R685A spike mutants for 1 hour, then control and antibody-treated PPs were used to infect HEK293T-ACE2 cells for 6 hours and further incubated for 48 hours (**Fig 4A**). We observed no significant difference in CB6-induced 50% neutralization dose ($ND_{50}$) (1.156 nM (0.037μg/mL) for WT PPs; 2.845 nM for R685A PPs) for PP infection of HEK293T-ACE2 cells (**Fig 4B**). Next, in the cell-cell fusion neutralization assays, we pre-incubated same doses of CB6 antibodies with spike-expressing HEK293T cells for 1 hour, before co-culturing with ACE2-expressing cells for 6 hours (**Fig 4C**). Interestingly, in this case we detected significant differences between CB6-induced neutralization against WT and R685A spike-expressing cells fusing with ACE2-expressing cells (**Fig 4D**).

We further validated the extent of S1 shedding in the cell-cell fusion model using immunoblotting. Indeed, at 100 nM, a dose approximately 50-folds higher than its reported $ND_{50}$ on PPs, we found that CB6 only marginally reduced ~20% cell-cell fusion between WT spike and ACE2-expressing cells (**Fig 4E, top left**). The spike R685A mutant, however, remained highly sensitive to CB6 neutralization when co-cultured with ACE2-expressing cells, and exhibited ~90% reduction in the bioluminescence signal (**Fig 4E, top right**). Supernatants collected from WT, but not R685A spike mutant, showed strong S1 shedding in the presence of ACE2 cells, or ACE2 cells with the CB6 neutralization antibody (**Fig 4E, bottom**); in contrast, S2' cleavage products collected from WT and R685A spike mutant cell lysates showed no difference. In supporting to CB6, neutralization capacity of REGN10933 at 100 nM was strongly improved in the R685A compared to WT spike, visualized by the ZsGreen fluorescent images (**S4A Fig**). Thus, in the absence of S1/S2 cleavage, CB6 could induce the internalization of R685A spike and prevent subsequent cell-cell fusion with ACE2-expressing cells.

Next, we utilized the furin inhibitor RVKR and tested whether inhibiting furin-mediated S1/S2 cleavage could restore the neutralization of WT spike by the CB6 antibody. We pre-incubated 100 nM CB6 with HEK293T cells expressing WT spike in the presence or absence of RVKR, before adding on HEK293T cells expressing ACE2. Compared to WT spike-mediated cell-cell fusion, 25 μM RVKR treatment significantly restored the neutralization capacity of CB6 in the cell-cell fusion model (**Fig 4F, top**), and efficiently prevented the S1 shedding induced by CB6 and ACE2-expressing cells (**Fig 4F, bottom**). Moreover, RVKR significantly improved CB6-induced neutralization of WT spike- and ACE2-mediated cell-cell fusion at various concentrations when compared to vehicle DMSO alone (**Fig 4G**), highly mirroring the neutralization effect of CB6 on the R685A spike mutants (**Fig 4E**). Restoration of CB6-mediated neutralization was also achieved using furin knockdown cells (**S4B Fig**), since inhibition

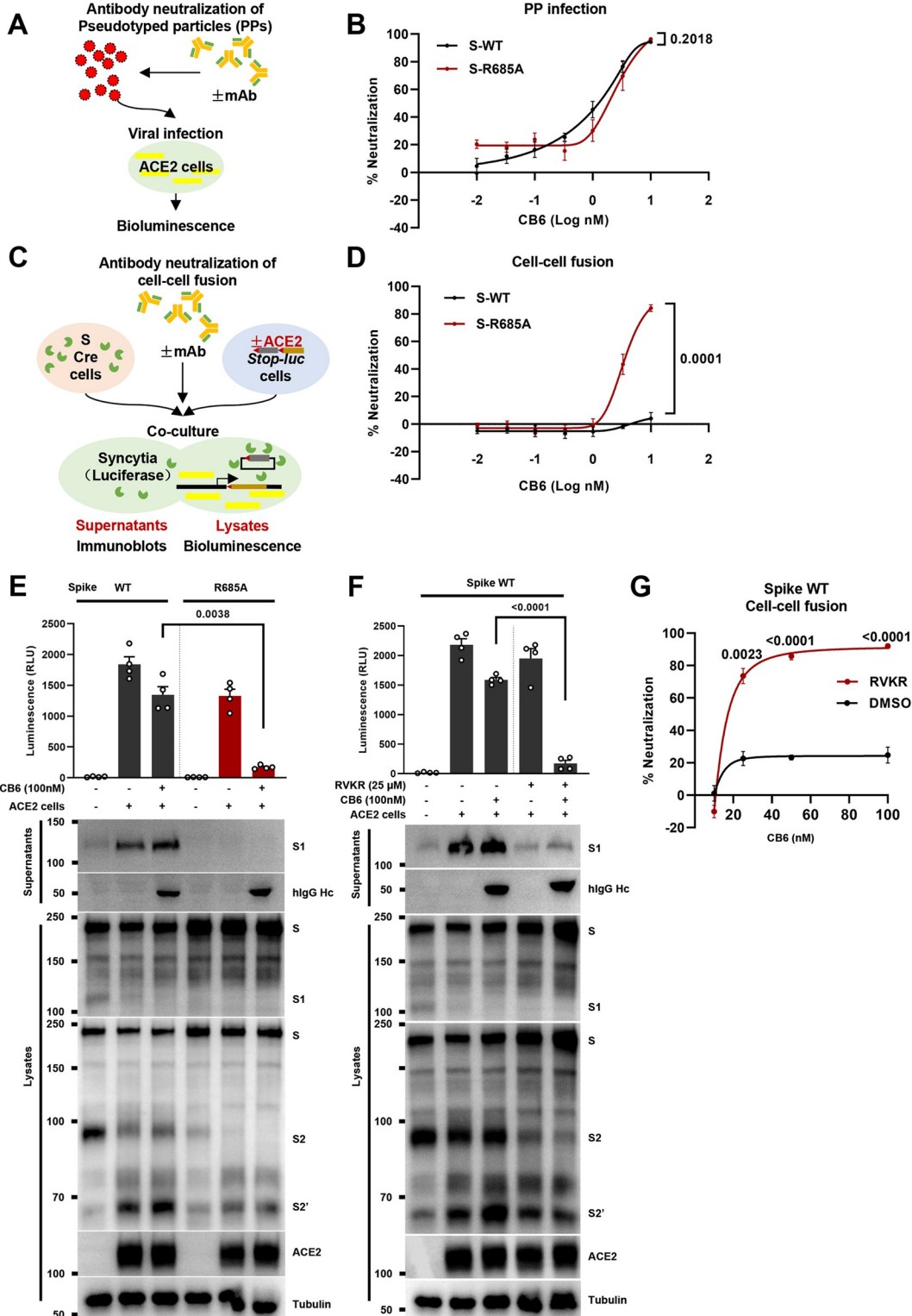

**Fig 4. Inhibition of S1 shedding restores the neutralization efficiency of RBM antibodies.** (**A**) Schematic representation of antibody neutralization of pseudotyped particles (PPs): serial diluted antibodies are pre-incubated with PPs prepared using SARS-CoV-2 spike and a retroviral transfer vector harboring a luciferase reporter for 1 hours at 37°C, before infection of ACE2-expressing cells for 48 hours at 37°C. Bioluminescence activity were quantified at end-point as the PP infection rate; (**B**) Neutralization curve of CB6 monoclonal antibody on PPs prepared using WT or R685A spike mutants. Percentage of

neutralization were normalized against the infection control pre-incubated with PBS, data are representative of six individual repeats; (**C**) Schematic representation of antibody neutralization of the cell-cell fusion: serial diluted antibodies are pre-incubated with HEK293T cells co-expressing SARS-CoV-2 spike and Cre for 1 hour at 37˚C, before co-cultured with HEK293T cells co-expressing control or ACE2 and *Stop-Luc* for 6 hours at 37˚C. Fusion-induced bioluminescence activity were quantified at end-point; (**D**) Neutralization curve of CB6 monoclonal antibody on WT or R685A spike-induced cell-cell fusion. Percentage of neutralization were normalized against the cell-cell fusion control pre-incubated with PBS, data are representative of six individual repeats; (**E**) Luciferase activity (RLU) measured from HEK293T cells co-expressing Cre and WT or R685A spike mutants without or with 100 nM CB6 pretreatment for 1 hour, then co-cultured with *Stop-Luc*-expressing control or ACE2 cells for further 6 hours (top); and immunoblots showing shedded S1 subunits, hIgG Hc, full-length spike, S1, S2, cleaved S2' and ACE2 collected from co-cultured cell supernatants and lysates (bottom). Blots are representative of four independent experiments; (**F**) Luciferase activity (RLU) measured from HEK293T cells co-expressing Cre and WT spike without or with 100 nM CB6 pretreatment in the presence of 25 μM RVKR, before mixing with *Stop-Luc*-expressing cells carrying control or ACE2 for further 6 hours (top); with immunoblots showing shedded S1 subunits, IgG Hc and full-length spike, S1, S2, cleaved S2' and ACE2 collected from co-cultured cell supernatants and lysates (bottom). Blots are representative of four independent experiments; (**G**) Percentage neutralization of increasing concentrations of (10, 25, 50 and 100 nM) CB6 for 1 hour, before co-cultured with *Stop-Luc*-expressing cells carrying ACE2 for further 6 hours. *P* values were obtained by one-way ANOVA with Sidak's *post hoc* test and are indicated on the figure.

of host cell furin prevented the CB6-induced activation of spike. In addition, RVKR potently reduced cell-cell fusion in the presence of CB6, as well as s1 shedding in both WT and Delta spikes (**S4C and S4D Fig**). These data suggest that antibody-mediated neutralization of cell-cell fusion can be restored when the spike proteolytic processing at the S1/S2 cleavage site is prevented.

## Protease-induced cleavage at S1/S2 bridge site promotes antibody evasion

Although R685A spike mutant lacking the furin cleavage motif was not processed into S1/S2 in HEK293T cells, arginine residues at R682 and R683 still render the R685A spike susceptible to other trypsin-like proteases, for instance TMPRSS2 (**Fig 5A**). Hence, we prepared non-cleavable spike ΔRRAR mutant as an additional mutant to test whether addition of exogenous trypsin would promote the activation of spike by extracellular RBM antibody. When 5 μg/mL trypsin was added to the supernatants of HEK293T cells expressing the R685A spike mutant for 6 hours, shedded S1 in the supernatant was not detectable, while spike in the cell lysates was readily processed into S1/S2 (**Fig 5B**), confirming our previous observation that trypsin alone is insufficient to trigger S1shedding; while the spike ΔRRAR mutant lacking the S1/S2 cleavage site did not process into S1/S2 when extracellular trypsin was added (**Fig 5B**). Based on these findings, we pre-incubated CB6 with HEK293T cells expressing the R685A spike mutant, in the absence or presence of trypsin, and re-assessed whether proteolytic cleavage at S1/S2 site by extracellular trypsin could promote antibody-mediated activation. Here we found that addition of 1 μg/mL trypsin enabled the shedding of S1 into the supernatants in the R685A spike supernatant stimulated with 12.5 nM CB6 (**Fig 5C, bottom**); addition of trypsin also restored cell-cell fusion stimulated by the CB6 as measured by a significant increase in the bioluminescence (**Fig 5C, top**). As a result, in the presence of extracellular trypsin, R685A spike mutant was no longer sensitive to 100 nM CB6-mediated neutralization when co-cultured with ACE2-expressing cells (**Fig 5D**). In contrast, spike ΔRRAR that deficient in S1/S2 cleavage site, was resistant to 1 μg/mL of trypsin pre-incubation and still sensitive to the same dose of CB6 antibody (**Fig 5E**). Thus, the proteolytic cleavage event at the S1/S2 bridge site modulates the spike activation switch on the plasma membrane, and could contribute to a mechanism of antibody evasion and prevents RBM antibody mediated neutralization in the cell-cell fusion.

## Antibody-mediated spike activation during SARS-CoV-2 infection

In order to validate the effect of antibody-mediated spike activation on cells during an authentic virus infection, we inoculated wildtype Vero E6 cells with 2 multiplicity of infection (MOI)

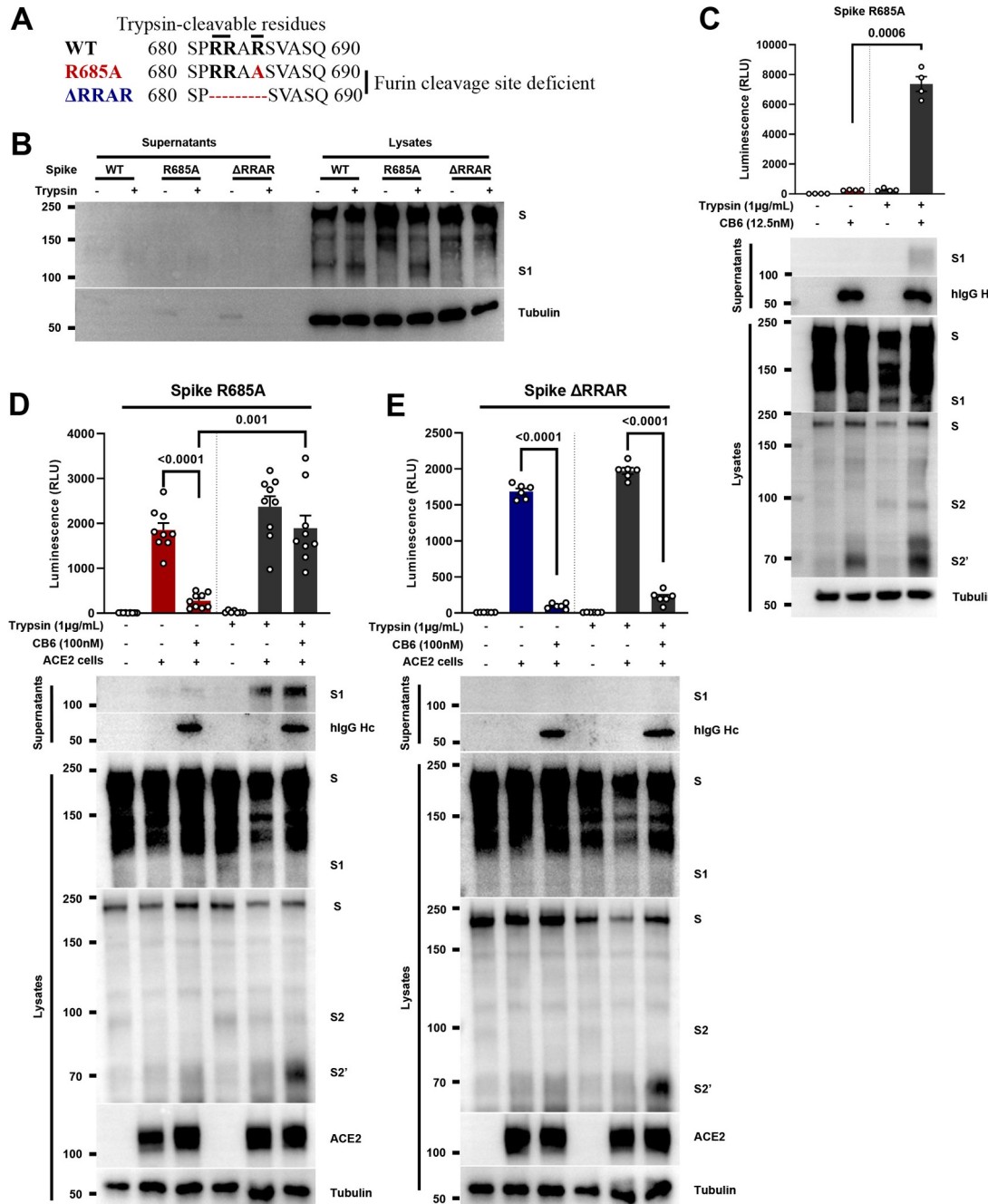

**Fig 5. Protease-induced cleavage at S1/S2 bridge site promotes antibody evasion.** (**A**) Amino acid sequence alignment of the SARS-CoV-2 WT, R685A and ΔRRAR spike mutants at S1/S2 cleavage site. Polybasic and trypsin-cleavable arginine residues are labeled in bold; (**B**) Immunoblots showing full-length spike and S1, collected from supernatant and lysate fractions of HEK293T cells expressing WT, R685A and ΔRRAR spike mutants treated without or with 5 μg/mL TPCK-trypsin for 6 hours. Blots are representative of two independent experiments; (**C**) Luciferase activity (RLU) measured from HEK293T cells co-expressing Cre and R685A spike mutants, stimulated with 12.5 nM CB6 without or with 1 μg/mL TPCK-trypsin for 16 hours (top); and immunoblots showing shedded S1 subunits, hIgG Hc and full-length spike, S1, S2 and cleaved S2' collected from co-cultured cell supernatants and lysates (bottom). Data and blots are representative of four independent experiments; (**D**) Luciferase activity (RLU) measured from HEK293T cells co-expressing Cre with R685A spike mutant, pretreated with 100 nM CB6 for 1 hour in the absence (Red bars) or presence (Gray bars) of 1 μg/mL trypsin, then co-cultured with *Stop-Luc*-expressing cells carrying control or ACE2 for further 6 hours (top); and immunoblots showing shedded S1 subunits, human hIgG Hc and full-length spike, S1, S2, S2' and ACE2 collected from co-cultured cell supernatants and lysates (bottom). Data are representative of nine individual repeats, blots are representative of six independent experiments; (**E**) Luciferase activity (RLU) measured from HEK293T cells co-

expressing Cre with ΔRRAR spike mutant, pretreated with 100 nM CB6 for 1 hour in the absence (Blue bars) or presence (Gray bars) of 1 μg/mL trypsin, then co-cultured with *Stop-Luc*-expressing cells carrying control or ACE2 for further 6 hours (top); and immunoblots showing shedded S1 subunits, human hIgG Hc and full-length spike, S1, S2, S2' and ACE2 collected from co-cultured cell supernatants and lysates (bottom). Data are representative of six individual repeats, blots are representative of two independent experiments. *P* values were obtained by one-way ANOVA with Sidak's *post hoc* test and are indicated on the figure.

live SARS-CoV-2 (isolate Wuhan-Hu-1) virus to establish the initial viral entry, then replaced growth medium without or with 12.5 or 25 nM CB6, and further incubated for 48 hours. Supernatants were eventually used for determining the viral titer derived by 50% tissue culture infectious dose ($TCID_{50}$), while cell lysates were used for the detection of SARS-CoV-2 spike and nucleocapsid (N) protein using immunoblots (**Fig 6A**). Although no noticeable change in cell morphology 24 hours post infection, SARS-CoV-2 induced extensive cytopathic effect (CPE) 48 hours after infection when compared to the mock control (**Fig 6B**). Of note, 12.5 or 25 nM CB6-treated cells developed significant amount of CPE with distinct patched and clustered morphology 48 hours post-infection (**Fig 6B**). When supernatants containing live SARS-CoV-2 virus and antibodies were further inoculated onto fresh Vero E6 cells to determine the infectious viral titer, 12.5 and 25 nM CB6 almost completely blocked SARS-CoV-2 further infecting Vero E6 cells (**Fig 6C**). But more importantly, when SARS-CoV-2 infected cells were immunoblotted for possible spike cleavage and potential increase in N, we detected robust cleavage of S2' in the infected group, and a concentration-dependent increase of S2' cleavage, as well as N protein, induced by the CB6 antibody (**Fig 6D**). These results suggested that the presence of CB6 antibody in supernatants had no effect on intracellular viral proteins and did not prevent the formation of CPE induced by the SARS-CoV-2 virus. On the contrary, presence of CB6 antibody could facilitated the spike proteolytic processing on infected cells. Hence, RBM neutralization antibody exerts differential effect on viral supernatants and infected cells.

Apart from immunoblotting cell lysates of SARS-CoV-2 infected Vero E6 cells, we also performed immunostaining on cells infected with SARS-CoV-2, treated without or with 12.5 nM CB6 antibody. At 48 hours post-infection, we detected robust co-staining of SARS-CoV-2 N positive and human IgG patches (reminiscent of CB6) on Vero E6 cells (**Fig 6E, top**); effect of CB6 on SARS-CoV-2 syncytia formation was further enhanced when SARS-CoV-2 infected Vero E6 cells over-expressing the TMPRSS2 (**Fig 6E, bottom**).

Since Vero E6 cells express endogenous ACE2 could promote the entry of SARS-CoV-2 and cell-cell transmission of the virus, we sought to validated the role of CB6 on Vero E6 cells using the anti-ACE2 blocking antibody after the initial viral attachment. By immunostaining SARS-CoV-2 N positive cells, we found that 1 μg/mL anti-ACE2 blocking antibody potently prevented majority of SARS-CoV-2 infection at 48 hours post-infection, limiting the initially infected cells as single colonies (**Fig 6G**). However, when both anti-ACE2 antibody and 12.5 nM CB6 were used, number of N positive cells was significantly increased in WT (**Fig 6G**), as well as the TMPRSS2 over-expressing Vero E6 cells (**S5A Fig**). Immunoblotting of these cell lysates confirmed that, presence of CB6 enhanced the expression of SARS-CoV-2 spike and N protein in the presence of anti-ACE2 blocking antibody (**Fig 6H**); and to a more extent in the Vero E6-TMPRSS2 cells (**S5B Fig**). These data confirmed that CB6 antibody drive cell-cell transmission of the SARS-CoV-2 under authentic infection conditions.

## Discussion

Majority of SARS-CoV-2 neutralization antibodies are currently being studies for their ability to bind RBD and block spike activation by various neutralization mechanisms. RBM

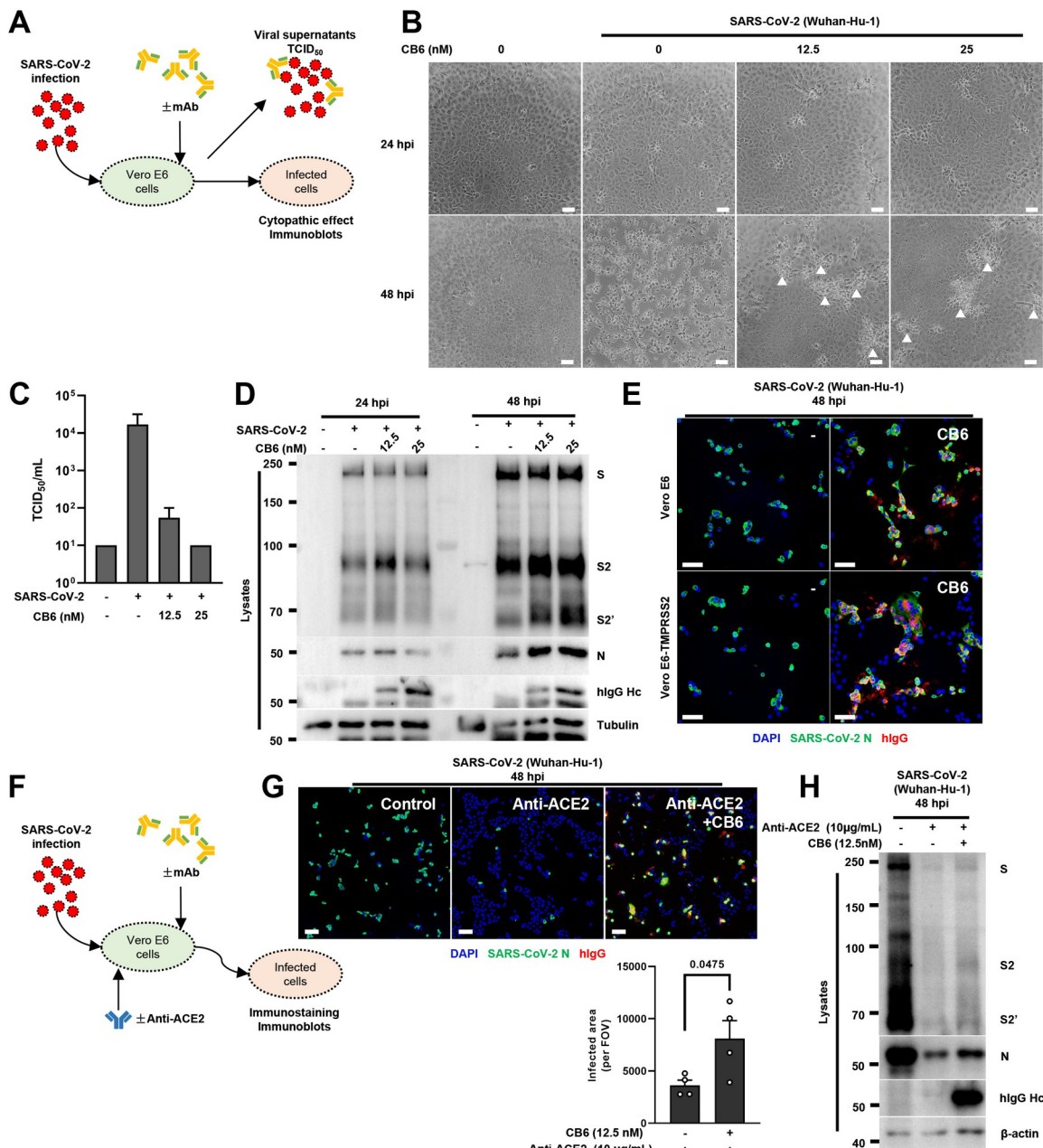

**Fig 6. Antibody-mediated spike activation during SARS-CoV-2 infection.** (**A**) Schematic representation of antibody-treatment on SARS-CoV-2 post-infected cells. Wildtype Vero E6 cells or Vero E6-TMPRSS2 cells were inoculated with 2 MOI live SARS-CoV-2 (isolate Wuhan-Hu-1) virus for 4 hours to ensure efficient viral entry; culture growth media were then replaced without or with antibody to investigate the possible role of antibody on post-infected cells. Supernatants and cell lysates were harvested 48 hours post infection (hpi) for viral titer and spike cleavage; (**B**) Brightfield images of 2 MOI SARS-CoV-2 infected wildtype Vero E6 cells, treated without or with 12.5 or 25 nM CB6 antibody for 24 and 48 hours. White arrows indicate patched cytopathic effect under the effect of antibody; scale bars are indicative of 50 μm and images are representative of two independent experiments; (**C**) SARS-CoV-2 viral titer measured on fresh Vero E6 cells by 50% tissue culture infectious dose (TCID$_{50}$), inoculated with serially diluted supernatants collected from (B). TCID$_{50}$ was measured with the Reed–Muench method, data were representative of two individual repeats; (**D**) Immunoblots of SARS-CoV-2 full-length spike, S2, S2' and N proteins, collected from wildtype Vero E6 cell lysates 24 and 48 hpi as described in (B). Blots are representative of three individual experiments; (**E**) Immunofluorescent images showing morphology of 2 MOI SARS-CoV-2 infected VeroE6 and VeroE6-TMPRSS2 cells, treated without or with 12.5 nM CB6 antibody 48 hpi. Anti-SARS-CoV-2 N and Anti-human IgG (H+L chains) were stained with Alexa fluor 488 and 555 respectively. Scale bars are indicative of 50 μm; (**F**) Schematic representation of anti-ACE2 blocking antibody and CB6 treatment on SARS-CoV-2 post-infected cells, wildtype Vero E6 cells were inoculated with 2 MOI live SARS-CoV-2 (isolate Wuhan-Hu-1) virus for 4 hours to before medium was replaced without or with 1 μg/mL anti-ACE2 blocking antibody and 12.5 nM CB6, cells were further incubated for 48 hours; (**G**) Immunofluorescent images showing

morphology of 2 MOI SARS-CoV-2 infected wildtype Vero E6 cells, treated without or with 12.5 nM CB6 antibody in the absence or presence of 1 μg/mL anti-ACE2 blocking antibody for 48 hpi. Anti-SARS-CoV-2 N and Anti-human IgG (H+L chains) were stained with Alexa fluor 488 and 555 respectively. Scale bars are indicative of 50 μm; (**H**) Immunoblots of SARS-CoV-2 full-length spike, S2, S2' and N, collected from wildtype Vero E6 lysates described in (G), blots are representative of two individual repeats.

antibodies binding to the virion spike neutralizes the SARS-CoV-2 virus by mimicking receptor ACE2 and converting spike into a post-fusion conformation [28,29,46]. We demonstrated that the monoclonal antibody CB6 has an efficient ND50 against WT and R685A spike PPs at nanomolar range (**Fig 4B**), agreeing the previously reported concentration. However, when soluble CB6 was used on SARS-CoV-2 spike-expressing cells, cell-cell fusion was robustly detected in the absence of ACE2 receptor. Among CB6 and REGN10933 antibodies, binding affinity to the specific RBM epitopes may affect the differences in terms of cell-cell fusion. Compared to other RBD neutralization antibodies, CB6 was capable of inducing the spike proteolytic activation at both S1/S2 and S2' cleavage sites on spike-expressing cells. Interestingly, CB6-induced cell-cell fusion was only present in the S1/S2 cleaved spike, since the S1 shedding enabled the activation of SARS-CoV-2 spike and the S2' cleavage at the plasma membrane (**Fig 7A**); while in the absence of S1/S2 cleavage, CB6 binding did not induce S1 shedding and promoted the internalization of the spike, hence, no cell-cell fusion was observed (**Fig 7B**).

Using extensive cellular models, here we demonstrated that soluble RBM antibodies binding to SARS-CoV-2 spike on cell surface, which triggered ACE2-independent cell-cell fusion among adjacent cells. Unlike ACE2, the lack of transmembrane domain in antibodies is unfavorable for tethering the membrane fusion process, since the opposing cell membrane is not physically pulled by an anchored receptor. Notably, activation of spike by soluble agonists has been previous described in murine hepatitis virus (MHV), where addition of a soluble form of carcinoembryonic antigen-related cell adhesion molecule 1 (CEACAM1) could increase syncytia formation among MHV infected cells [47]. Interestingly, MHV spike from JHMV strain carrying a polybasic S1/S2 cleavage site (RRARR) display enhanced cell-cell spread when compared to the MHV-A59 strain where S1/S2 cleavage site is RRAHR [48,49]. Other coronavirus spike such as MERS-CoV [27], human coronavirus OC43 spike that carry polybasic cleavage site at the S1/S2 junction could also share a similar mechanism; in contrast, SARS-CoV spike carrying monobasic S1/S2 cleavage sites could be subjected to additional cleavage by exogenous proteases such as TMPRSS11D and TMPRSS2 [50,51], in addition to host cell furin. Whether strain specific RBM antibodies could also drive cell-cell fusion of the other coronavirus spike remain to be determined in the future.

Mechanistically, the detachment of S1 may serve to accelerate the uncapping of the fusogenic S2 subunit, while enable the insertion of fusion peptide into nearby plasma membranes to initiate the ACE2-independent cell-cell fusion [17,52–54]. It is unclear whether other associated factors, such as lectins, NRP1 and TMEM106B [23,24,55,56], or glycosylation profiles of adjacent residues near the S1/S2 cleavage site [57], affect spike activation and cell-cell fusion. In addition to soluble agonists of spike, bound antibodies can be recognized by the Fc receptors expressed by immune cells, where spike-mediated membrane fusion could contribute to the cell-in-cell formation as previously reported [58].

Unlike free viral particles, syncytia formation allows the transmission of viral contents between cells without encountering extracellular environment, where soluble antibodies are highly enriched. Others have reported that spike-mediated cell-cell fusion among ACE2 cells is highly resistant to neutralization antibodies as well as convalescent and vaccinated sera [31,32,55,59]. In several animal models, presence of the furin cleavage site on the SARS-CoV-2 spike is strongly associated with the lung pathogenesis *in vivo* [15,21], whether serum-derived

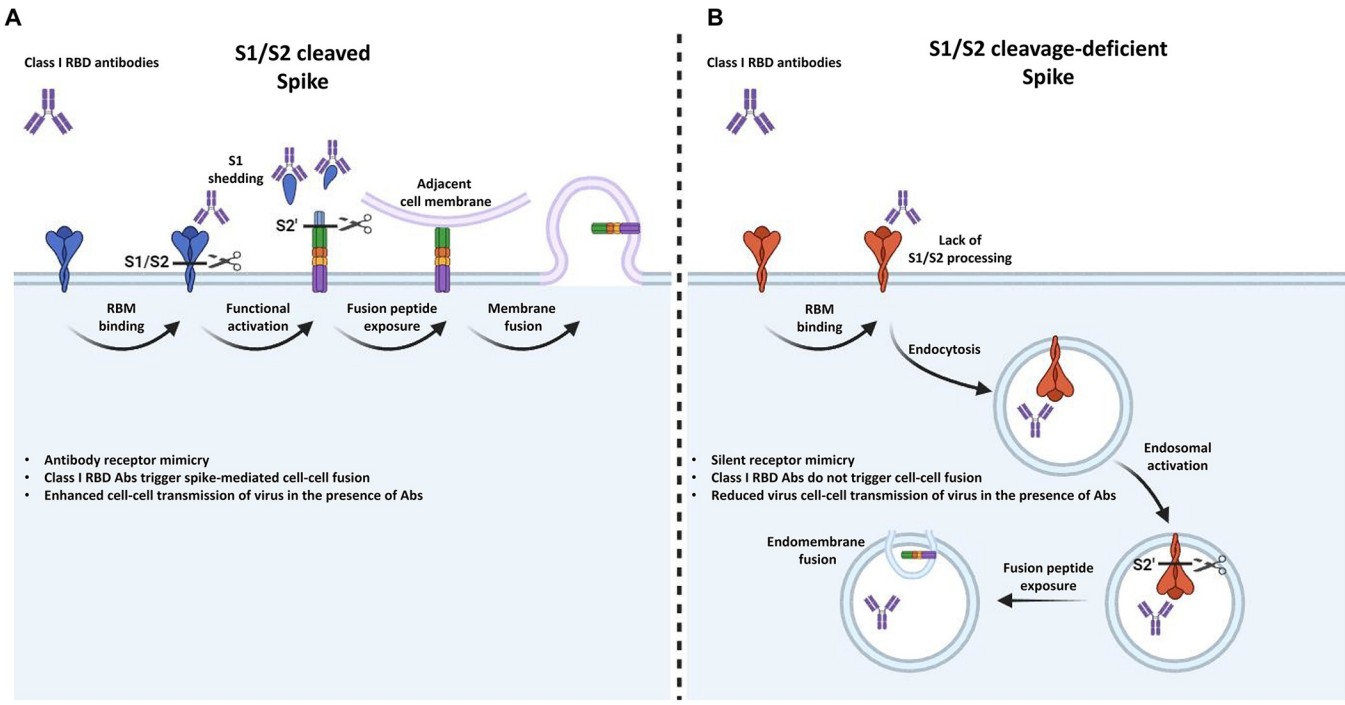

**Fig 7. Graphical abstract of antibody-mediated spike activation.** (**A**) When SARS-CoV-2 spike is cleaved at the S1/S2 cleavage site and expressed on the infected cell membrane, binding of Class I antibodies (Abs) onto the receptor binding motif (RBM) trigger the rapid shedding of S1 subunit at the cell surface. This event allows the functional activation of the S2' cleavage site by membrane bound proteases, such as TMPRSS2. Exposure of fusion peptide at the plasma membrane triggers receptor-independent cell-cell fusion among adjacent cells. This process drives the functional activation of spike-expressed on the cell membrane and promote cell-cell transmission of the SARS-CoV-2 virus; (**B**) However, when spike S1/S2 site is not cleaved by an endogenously expressed protease, for instance by host cell furin or TMPRSS2, binding of Class I Abs on the RBM is unable to trigger S1 shedding from the spike-expressing cells. Instead, binding of Class I Ab leads to the internalization of spike trimers, where S2' cleavage and exposure of fusion peptide could occur inside endolysosomes. As a result, cell-cell transmission of the SARS-CoV-2 virus could be efficiently prevented. Figure was illustrated using images created with BioRender.com.

RBM antibodies could exacerbate the pathology on infected tissues remain an area of interest. When compared to the SARS-CoV and bat-related coronaviruses, presence of polybasic cleavage in SARS-CoV-2 spike may provide an evolutionary advantage in transmitting among infected tissue, likely to prolong the viral replication cycle.

## Limitation of the study

Although we tested spike activation in response to individual classes of monoclonal antibodies *in vitro*, it is unlikely to account for a polyclonal humoral response complementing multiple spike epitopes during an authentic viral infection. For instance, non-RBM antibodies, such as fusion peptide and stem helix antibodies [60,61] may prevent antibody-mediated spike activation downstream of receptor mimicry. Antibody cocktails and patient plasma simultaneously targeting distinct epitopes along the spike trimer should be used to extend the scope of this study. Although cell-cell fusion among pneumocytes have a strong implication for human lung pathology, it is unclear whether specific RBM antibodies could exacerbate the disease progress *in vivo*, especially in the polarized airway epithelial cell models. Timing of the monoclonal therapeutic antibody treatment could be important, and more studies are required to indicate appropriate antibodies for future prophylactic or therapeutic purposes.

Shedding of viral glycoprotein is common among other Class I fusion proteins. Most notably the shedding of GP120 from HIV-1 envelop (GP160) [62]. It is currently unclear whether shedding of GP120 could promote antibody insensitivity towards patient anti-sera [63,64].

Presence of furin-cleavage site on West Nile Viruses envelop proteins have also been reported to alter the neutralization response [65]. For SARS-CoV-2, human monoclonal antibody targeting epitopes adjacent to the polybasic cleavage site has yet to be identified, such candidate should be tested against the S1/S2 cleavage event, as well as the synergic activity with other neutralization antibodies. In summary, we demonstrated an important aspect of RBM antibodies on SARS-CoV-2 infected cells displaying the spike, highlighting an important role for the coronavirus spike S1/S2 cleavage site, and revealed a novel target for the prevention of coronavirus cell-cell transmission.

## Materials and methods

### Ethics statement

No animal experiment was conducted for this study. All experiments involving live SARS-CoV-2 virus were conducted in the biosafety level 3 (BSL3) laboratory of the Shanghai municipal center for disease control and prevention (CDC). All SARS-CoV-2 isolates and viral stocks were sequenced before use; experiments and protocols in this study were screened and approved by the Ethical Review Committee of the Shanghai CDC.

### Cell culture and purification of antibodies

All cell lines were obtained from the National Science & Technology Infrastructure (NSTI) cell bank (www.cellbank.org.cn). Vero E6 cells, HEK293T cells and HEK293T stably expressing human ACE2 (HEK293T-ACE2) cells were cultured in Gibco Dulbecco's Modified Eagle Medium (DMEM) (GE Healthcare) supplemented with 10% fetal bovine serum (FBS) (Sigma) and 1% Penicillin/streptomycin (P/S) (Life Technologies) at 37°C with 5% $CO_2$ in a humidified incubator. Human lung cancer cell line Calu-3 cells were cultured in Minimum Essential Medium (MEM) supplemented with 10% FBS, 1% non-essential amino acids and 1% P/S (Procell). All cells were routinely tested for mycoplasma contamination and passages between 4[th] to 20[th] were used.

Human monoclonal antibodies were purified using a protocol previously described [40]. Briefly, DNA encoding the CB6, FD20, REGN10933 and REGN10987 heavy and light chain variable regions was separately Gibson-assembled into a pDEC vector which contains the human immunoglobulin (IgG) backbone. IgG plasmids were transiently transfected into Expi293 cells (A14527, Thermo Fisher) using polyethylenimine at a cell density of $2 \times 10^6$ per milliliter. Culture medium containing secreted IgGs were filtered and immunoprecipitated with Protein A beads. Purified monoclonal antibodies were then washed and eluted with low pH Tris-glycine buffer before dialyzed and desalted with phosphate buffered saline (PBS) on a desalting column (Biorad).

### Reagents and plasmids

Bovine pancreatic TPCK-treated trypsin (PI20233) was purchased from Thermo Fisher; Dec-RVKR-cmk (HY-107760) and Bafilomycin A1 (HY-100558) was purchased from MedChemExpress. ACE2 polyclonal antibody (21115-1-AP) was purchased from Proteintech. Rabbit anti-SARS-CoV-2 S2 (40590-T62) polyclonal antibody was used for the detection of S2', S2 and S proteins; mouse anti-SARS-CoV-2 N (40143-MM05) monoclonal antibody was used for the detection of nucleocapsid proteins and mouse monoclonal anti-S1 (40591-MM42) and (40592-MM117, Omicron-specific) for the detection of shedded S1 protein were purchased from Sino Biological. Rabbit anti-PARP antibody (9532S), anti-EEA1(3288S) and rabbit anti-LAMP1 (9091S) were purchased from CST, and rabbit anti-human furin polyclonal antibody

(18413-1-AP) was purchased from Proteintech. The mouse monoclonal anti-human ACE2 blocking antibody was purchased from Sino Biological (10108-MM37). β-tubulin was used as loading control and the mouse anti-β-tubulin (AC030) monoclonal antibody was purchased from Abclonal. Horseradish Peroxidase (HRP) conjugated goat anti-mouse (115-035-003, Jackson ImmunoResearch), anti-rabbit (111-035-003, Jackson ImmunoResearch) and goat anti-human Fc specific (A0171, Sigma) secondary antibodies were used. For immunostaining, goat anti-Human IgG (H+L)-555 (A48277), goat anti-rabbit IgG (H+L)-488 (A11034) and goat anti-mouse IgG (H+L)-488 (A11029) were purchased from Invitrogen.

SARS-CoV-2 spike (Wuhan-Hu-1, GenBank: QHD43419.1) was homo sapiens codon-optimized and generated *de novo* into pVAX1 vector by recursive polymerase chain reaction (PCR). Spike WT, Alpha, Beta and Delta variants containing point and deletion mutations were generated using stepwise mutagenesis using spike construct containing the truncated 19 amino acids at the C-terminal (CTΔ19). The latest human codon optimized Omicron BA.1 and BA.4 plasmids were purchased from Genescripts (MC_0101274), and subcloned into the pVAX1 backbone with CTΔ19 for comparison. Plasmids encoding Cre recombinase and the floxP stop-luciferase were kindly provided by Prof Jin Zhong; For visualizing fused cells, the luciferase gene was modified and replaced with Discosoma sea anemones mCherry. Human ACE2 assembled in a pcDNA4.0 vector was used for transient expression of ACE2. Site-directed mutagenesis and deletions were performed using customized primers (synthesized by Sangon and Biosun) and KOD PLUS (TOYOBO) high fidelity polymerase. Parental methylated DNA was digested using Dpn I and the resultant PCR products were then transformed into XL10-Gold ultra-competent E. coli (Agilent). Plasmids were then extracted using DNA extraction miniprep kits and DNA concentrations were adjusted using Nanodrop (Thermo). All mutant plasmids were validated by sequencing service provided by Sangon and Biosun.

## Transient transfection and cell-cell fusion assays

For transient transfections, HEK293T cells were seeded in flat bottom 24-well plates at $0.5 \times 10^6$ cells /mL density overnight. 250 ng plasmids encoding SARS-CoV-2 spike mutants or ACE2 variants were packaged in Lipofectamine 2000 (Life technologies) and transfected for 24 hours. To quantify spike-mediated membrane fusion, a *Cre-loxp* Firefly luciferase (*Stop-Luc*) co-expression system was introduced to enable the detection of DNA recombination events during cell-cell fusion as previously described [66]. 200 ng Cre plasmids were co-transfected into S-HEK293T cells and 200 ng *Stop-Luc* plasmid were co-transfected into ±ACE2-- HEK293T cells, respectively. For visualization of syncytia formation, 100 ng ZsGreen plasmids were co-transfected with spike variants; 200 ng *stop-mCherry* plasmids were transfected into control HEK293T cells for the visualization of the antibody-induced cell-cell fusion. HEK293T cells in the 24-well plates were then detached using ice-cold calcium-free phosphate buffered saline (PBS) in the absence of trypsin and centrifuged at 600 g for 4 min.

For spike-mediated cell-cell fusion assays, cell pellets were resuspended into complete DMEM and mixed with control HEK293T cells, or HEK293T cells expressing ACE2 and HEK293T-ACE2 cells at 1:1 ratio before adhesion to the 48-well or 96-well plates, cell mixes were incubated for 16 hours at 37˚C. Quantification of cell-cell fusion was performed by measuring luciferase expression as relative luminescence units (RLU) by mixing cell lysates with the Steady-Glo luciferase substrate (E2520, Promega) on a Synergy H1 plate reader (Biotek). For validation of cell viability of fused cells, cell lysates were subjected to the ATP CellTiter-Lumi Steady (Beyotime) luminescent cell viability assay according to the manufacturer's instructions.

Fluorescent images showing syncytia formation post ACE2- or CB6-treatment were captured at endpoint using a 20x objective and 12-bit monochrome CMOS camera installed on

the IX73 inverted microscope (Olympus). Attached cells and syncytia were lysed in a NP-40 lysis buffer containing 0.5% (v/v) NP-40, 25 mM Tris pH 7.3, 150 mM NaCl, 5% glycerol and 1x EDTA-free protease inhibitor cocktail (PIC) (Roche). For protease-induced cleavage of S1 protein, HEK293T expressing spike mutants were resuspended in serum-free DMEM containing 1% P/S without or with 1 μg/mL TPCK-treated trypsin (PI20233, Thermo Scientific) for 6 hours. For proteinase K digestion of spike, antibody-treated cell lysates were incubated with 10 μg/mL proteinase K (AM2548, Invitrogen) for 30 min at 37˚C, reaction was terminated by adding 5x reducing Laemmli loading buffer.

## Preparation of spike-expressing lentivirus and transduction

Spike-expressing lentivirus were packaged and produced in WT HEK293T cells. Briefly, plasmids encoding full-length WT spike were subcloned onto a Fugw backbone and co-transfected with plasmids encoding ZsGreen, pDelta8.9 and the VSV-G envelope glycoprotein at a 3:1:4:1 ratio into HEK293T cells. Growth medium were replaced 6 hours after transfection, and further incubated for 48 hours. Spike-lentivirus in the culture supernatants were then collected and filtered through a 0.22 μm PES filter, before used for lentiviral transduction.

## Pseudotyped particles (PPs) preparation and infection

Retroviral pseudotyped particles were packaged by co-transfection of HEK293T cells using plasmids encoding SARS-CoV-2 spike, murine leukemia virus (MLV) core/packaging components (Gag-Pol) and a retroviral transfer vector harboring a luciferase reporter, at a ratio of 1:5:3. Growth medium containing pseudotyped particles were harvested 48 h post-transfection and filtered through a 0.22 μm membrane before PP infection, neutralization assays and immunoblotting.

PP infection was performed using HEK293T without or with ACE2 expression, seeded at 5000 cells-per-well density in a 96-well plate. After 6 h of PP infection in the absence or presence of 50 nM bafilomycin A1 (MCE) or supernatants containing shedded S1, supernatants were removed and cells were further incubated in fresh DMEM containing 2% FBS and 1% P/S for 48 h at 37 ˚C. PP infectivity were quantified by measuring luciferase activity using Steady-Glo luciferase substrate. For immunoblotting analysis of the cleaved virion spike proteins, PP infections were performed for 16 h, and the infected cells were washed once in ice-cold $Ca^{2+}$-containing PBS, before collected and directly boiled in 2x reducing Laemmli buffer for immunoblotting.

## Immunoblotting and immunoprecipitation

Supernatants containing shedded S1 and cell lysates containing full-length spike were collected separately. Supernatants were directly boiled in the 5x Laemmli loading buffer, or used for subsequent immunoprecipitations. Tissue culture plates containing adherent syncytia and cell mixes were directly lysed on ice in 2x reducing Laemmli loading buffer before boiled at 95˚C for 5 min. Protein samples were separated by standard 7.5% or 9.5% Tris-glycine sodium dodecyl sulfate polyacrylamide gel electrophoresis (SDS-PAGE). Proteins were then transferred onto 0.45 μm PVDF membranes (Millipore) for wet transfer using Towbin transfer buffer. All membranes were blocked in PBS supplemented with 0.1% Tween20 (PBST) and 2.5% bovine serum albumin (BSA) or 5% non-fat dry milk, before incubation in primary antibodies. Blots were labelled with HRP-tagged secondary antibodies (Jackson ImmnuoResearch) and visualized with PicoLight substrate enhanced chemiluminescence (ECL) solution (Epizyme Scientific). Immunoblot images were captured digitally using a 5200 chemiluminescent imaging system (Tanon) with stacked molecular weight markers as indicated.

For co-immunoprecipitation, transiently transfected HEK293T cells were lysed on ice for at least 20 min in a NP-40 lysis buffer containing 0.5% (v/v) NP-40, 25 mM Tris pH 7.3, 150 mM NaCl, 5% glycerol and 1x EDTA-free protease inhibitor cocktail (Roche). WT or Delta spike-expressing cell lysates were pre-cleared with Protein A/G magnetic beads (B23202, Bimake) in the absence of antibody. Input samples were obtained from pre-cleared lysate mixes, and resultant lysates were then incubated with CB6 antibodies overnight at 4°C. Pull-down samples on the protein A/G magnetic beads were washed 3 times in NP-40 lysis buffer before boiling in 2x Laemmli loading buffer. Proteins were then detected by immunoblotting.

## Antibody neutralization assays

For neutralization of MLV pseudotyped particles, HEK293T-ACE2 cells were seeded at 5,000 cells-per-well in 96-well plates. Before PP infection, half-logarithmic dilutions (7 points, 0.1 nM to 10 nM) of CB6 monoclonal antibodies were pre-incubated with 100 μL PPs prepared with WT or R685A spike for 1 h at 37 °C; positive control PPs were incubated in the absence of neutralization antibody. HEK293T cells were then infected with PPs for 6 hours before further incubated for 48 h in fresh DMEM containing 2% FBS and 1% P/S. PP infectivity were quantified by measuring luciferase activity using Steady-Glo luciferase substrate, antibody neutralization activity was calculated by normalizing to the positive control PPs.

For the neutralization of cell-cell fusion, HEK293T cells co-expressing WT or R685A spike mutants with Cre recombinase were seeded in 96-well plates and pre-incubated with half-logarithmic dilutions of CB6 monoclonal antibodies for 1 h; HEK293T cells co-expressing control or ACE2 with *Stop-Luc* were then added and further incubated for 6 h; S-mediated cell-cell fusion against ACE2-expressing cells were used as positive control. Cells were subsequently lysed in NP-40 lysis buffer and luciferase activity were measured using the Steady-Glo luciferase substrate. Antibody neutralization was calculated and normalized against the cell-cell fusion positive control.

## RNA interference

Small interference RNA (siRNA) targeting the human furin (*siFURIN*) or non-targeting scrambling control (*siControl*) were synthesized by GenePharma. Three 21 nucleotide oligomers *siFURIN* (GeneID 5045), with sense (5'– 3') sequence GGACUUGGCAGGCAAU UAUTT, CUCCGCAGAUGGGUUUAAUTT, GGACCGCCUUUAUCAAAGATT were prepared as a mix; *siControl* with sequence UUCUCCGAACGUGUCACGUTT was used as a transfection control, *siControl* and *siFURIN* were prepared to a final concentration of 50 nM before transiently co-transfected with spike or ACE2 into HEK293T cells using Lipofectamine 2000 for 24 hours at 37°C. HEK293T cells with *siControl* and *siFURIN* knockdown were detached with cold PBS, and validated using immunobloting, before seeded in 96-well plate for the co-culture or antibody stimulations.

## Immunostaining and confocal microscopy

For immunostaining of the spike and antibodies, cells were fixed with 4% paraformaldehyde for 15 min, before being permeabilized with PBS containing 0.1% saponin for 5 min at room temperature. After blocking coverslips with 1% PBS for 30 min, rabbit anti-S2 or anti-LAMP1, primary antibodies were diluted 1:200 in the blocking solution and incubated with the coverslip for 1 hour. Coverslips were then washed 3 times with PBS, before stained with goat anti-human IgG (H+L)-555 (Invitrogen, A48277) Alexa Fluor secondary antibodies. Nuclei were subsequently stained with 4′,6-diamidino-2-phenylindole (DAPI) and mounted onto micro-slides using Vectashield antifade mounting reagent (Vectorlabs) and left to dry overnight at

room temperature. Fluorescent images covering various areas on the coverslips were randomly captured at 14-bit depth in monochrome using a 100x oil immersion objective on the Olympus SpinSR10 confocal microscope and subsequently processed using imageJ software (NIH) with indicated scale bars. Colocalization of human IgG with EEA1 was calculated as R values derived from Pearson's coefficients per field of view by using Image Pro Plus 6 (MediaCybernatics).

## Cell infection with authentic SARS-CoV-2 virus

Briefly, 1 day before the infection, wildtype Vero E6 or Vero E6-TMPRSS2 cells were seeded into 96-well or 24-well plates at a density of 4 x $10^5$ cells per mL, 2 multiplicity of infection (MOI) live SARS-CoV-2 was inoculated on to Vero E6 cells for 4 hours, before replaced with medium containing PBS control, or 12.5 and 25 nM CB6, without or with 10 μg/mL mouse anti-ACE2 blocking antibody, and further incubated for 24 and 48 hours. Brightfield images were captured to indicate the development of CPE and SARS-CoV-2 cell-cell transmission, cell lysates were collected for spike S2' cleavage and N protein immunoblots. In order to measure the virus-to-cell transmission and the 50% tissue culture infectious dose per milliliter (TCID50/mL), fresh Vero E6 were used and inoculated with serially diluted stock or antibody-treated SARS-CoV-2 virus supernatants for 1 hour, before washed and further incubated at 37 ˚C for 7 days. Cytopathic effect (CPE) appearance in the 96-well plate was observed by light microscopy and the TCID50/mL was calculated using the Reed–Muench method.

## Packaging of lentivirus and generation of CRISPR-Cas9 knockout cell lines

Briefly, sgRNA were cloned into LentiCRISPR v2 (#52961; Addgene) and packaged into lentiviral particles in HEK293T cells. The ACE2 sgRNA 5–3' sequences used were: TATGTTT-CATCATGGGGCAC. For packaging of lentivirus, HEK293T cells were seeded overnight, before co-transfected with a LentiCRISPR.v2 plasmid, psPAX2, and VSV-glycoprotein at a 4:3:2 ratio using Lipofectamine 2000 (11668019; Thermo Fisher Scientific). The supernatants containing *sgControl* and *sgACE2* lentiviral particles were harvested after 48 hours to transduce HEK293T and A549 cells. One day post-transduction, cells were subjected to puromycin selection at a concentration of 2 μg/ml for 72 h. Surviving cells were subjected to diluted to single cells and seeded in 96-well plates to obtain stable single clones deficient in the *ACE2* genes. Genomic DNA from single clones were then extracted amplified by PCR and sequenced to validate CRISPR-Cas9 knockout using (Forward CAGATTCCCCTGAAACTTTTTG and Reverse GGAGGTCTGAACATCATCAGTG) PCR primers.

## Statistics analysis

Bar charts were presented as mean values ± standard error of mean (SEM) with individual data points. All statistical analyses were carried out with the Prism software v8.0.2 (GraphPad). Data with multiple groups were analyzed using matched one-way ANOVA followed by Sidak's *post hoc* multiple comparisons. Statistical significance *P* values were directly indicated between compared groups and shown on figures.

## Supporting information

**S1 Fig. Characterization of antibody-induced cell-cell fusion.** (**A**) Immunoblot of full-length PARP, collected from HEK293T cell expressing WT spike, stimulated by 12.5 nM CB6, FD20, REGN10933 and REGN10987 for 16 hours. Blot is representative of two individual experiments; (**B**) Schematics of antibody-induced cell-cell fusion among cells co-expressing spike

and ZsGreen (Crosslinking syncytia, Green), or spike-to-adjacent cells expressing mCherry (Alternative syncytia, Yellow); representative images and quantification of crosslinked and alternative syncytia per field of view (FOV), stimulated without or with 12.5 nM CB6 for 16 hours, scale bars are representative of 50 μm. Data are representative of six individual repeats; (**C**) Immunoblots showing CB6 immunoprecipitants and input control of WT, Delta and Omicron BA.1 spike VOCs expressed in HEK293T cells. Blots are representative of two individual experiments; (**D**) Representative fluorescent images captured at 488 nm from HEK293T cells co-expressing MLV-gag, MLV-luc/ZsGreen, WT or Delta Spike VOCs, stimulated without or with 12.5 nM CB6 for 24 hours, scale bars are representative of 50 μm. Images are representative of two independent experiments, syncytia are indicated with white arrows; (**E**) Luciferase activity collected from HEK293T-ACE2 cells, after transduction by MLV-S PPs containing 12.5 nM CB6 prepared from (D). Data are representative of four individual repeats and displayed as individual points with mean ± standard error of the mean (SEM). *P* value was obtained by one-way ANOVA with Sidak's *post hoc* test and is indicated on the figure. (TIF)

**S2 Fig. Antibody-mediated cell-cell fusion is ACE2-independent.** (**A**) Luciferase activity (RLU) measured from cell-cell fusion assay and immunoblots showing shedded S1 subunits, hIgG Hc, full-length spike, S1, S2 and cleaved S2' collected from supernatant and cell lysate fractions of HEK293T cells stimulated with 0.1, 0.5, 2.5, 12.5 nM REGN10933 or Isotype Control for 16 hours. Data and blots were representative of four individual repeats; (**B**) Luciferase activity (RLU) measured from cell-cell fusion assay and immunoblots showing shedded S1 subunits, hIgG Hc, full-length spike, S1, S2 and cleaved S2' collected from supernatant and cell lysate fractions collected of CB6-stimulated HEK293T cells, treated without or with 2.5 μM EK1C4 for 16 hours. Data and blots are representative of five individual repeats; (**C**) Fluorescent images of mCherry reporter cell-cell fusion assay, captured at 594 nM from HEK293T cells co-expressing Beta N417 and K417 (reversion) with Cre, mixed with *stop-mCherry* HEK293T cells and stimulated without or with 12.5 nM CB6 for 16 hours, scale bars are representative of 50 μm; (**D**) Infectivity in RLU obtained from HEK293T-ACE2 cell lysates infected with SARS-CoV-2 MLV-S-WT pseudotyped particles in the presence of 0.01, 0.1, 1 and 10 μg/ mL mouse anti-ACE2 blocking antibody. Data are representative of four repeats; (**E**) Luciferase activity (RLU) measured from cell-cell fusion assay and immunoblots showing shedded S1 subunits, anti-ACE2 mIgG, hIgG Hc, full-length spike, S1, S2 and cleaved S2' collected from supernatant and cell lysate fractions of HEK293T cells stimulated with 12.5 nM CB6 antibody, without or with 1 μg/mL anti-ACE2 blocking antibody; (**F**) Luciferase activity (RLU) measured from cell-cell fusion assay and immunoblots showing shedded S1 subunits, hIgG Hc, full-length spike, S1, S2 and cleaved S2' collected from supernatant and cell lysate fractions of *sgControl* or *sgACE2* HEK293T cells stimulated with 12.5 nM CB6 antibody for 16 hours. Data are representative of three individual repeats; (**G**) Immunoblots showing shedded S1 subunits, anti-ACE2 mIgG, full-length spike and S1 collected from supernatant and cell lysate fractions of lentivirus transduced A549 cells expressing WT spike, stimulated with 12.5 nM CB6 antibody without or with anti-ACE2 blocking antibody for 16 hours. Blots are representative of two individual repeats; (**H**) Immunoblots showing shedded S1 subunits, hIgG Hc, full-length spike and S1 collected from supernatant and cell lysate fractions of lentivirus transduced *sgControl* and *sgACE2* A549 cells expressing WT spike, stimulated with 12.5 nM CB6 antibody. Blots are representative of two individual repeats. (TIF)

**S3 Fig. Antibody-mediated cell-cell fusion requires the furin cleavage of spike.** (**A**) Immunoblots showing proteinase K-resistant S2' cleavage product, obtained from WT and R685A

spike HEK293T cell lysates, stimulated with 12.5 nM CB6 for 16 hours. Lysates were then treated without or with 10 μg/mL proteinase K for 30 min at 37°C. Blots are representative of two individual repeats; (**B**) Representative confocal images of 12.5 nM CB6 antibody-stimulated HEK293T cells expressing WT or R685A spike mutant for 16 hours. Anti-LAMP1 and Anti-human IgG (H+L chains) were stained with Alexa fluor 488 and 555 respectively, co-localizations are indicated with white arrows, scale bars are representative of 10 μm. Images are representative of two individual experiments; (**C**) Immunoblots showing full-length S, cleaved S1, furin and tubulin, collected from HEK293T cells co-expressing 50 nM non-targeting (*siControl*) or human furin-targeting (*siFURIN*) siRNAs with full-length spike for 24 hours. Blots are representative of three individual repeats; (**D**) Representative fluorescent images of HEK293T cells co-expressing WT or Delta spike VOCs, co-cultured with pcDNA4 control or FURIN over-expressing cells stimulated without or with 12.5 nM CB6 antibody for 16 hours, scale bars are representative of 50 μm. Images are representative of two individual experiments; (**E**) Luciferase activity (RLU) measured from cell-cell fusion assay and immunoblots showing shedded S1 subunits, hIgG Hc, full-length spike, S1, S2 and cleaved S2' collected from supernatant and cell lysate fractions of HEK293T cells expressing spike WT or Delta VOC stimulated with 12.5 nM CB6 antibody for 16 hours. Data are representative of five repeats.
(TIF)

**S4 Fig. Dose-dependent neutralization of cell-cell fusion by CB6 in the R685A spike mutant.** (**A**) Fluorescent images of REGN10933-mediated neutralization of cell-cell fusion in HEK293T cells co-expressing ZsGreen reporter with WT or R685A spike mutant, preincubated without or with 100 nM REGN10933 for 1 hour, then co-cultured with HEK293T cells expressing ACE2 for 16 hours. Scale bars are indicative of 50 μm and images are representative of two independent experiments; (**B**) Luciferase activity (RLU) measured from HEK293T cells co-expressing Cre, *siControl* or *siFURIN* with WT spike pretreated with 100 nM CB6 for 1 hour, before mixing with *Stop-Luc*-expressing cells carrying ACE2 for further 6 hours (top); and immunoblots showing shedded S1, hIgG Hc, full-length spike, S2 and cleaved S2' collected from stimulated cell supernatants and lysates (bottom). Data are representative of four individual repeats, blots are representative of two independent experiments; (**C**) Immunoblots showing shedded S1 subunit and hIgG IgG Hc, full-length spike, S1, S2 and cleaved S2' collected from supernatants and lysates of HEK293T cells co-expressing ZsGreen reporter with WT and Delta spike VOCs, preincubated without or with 100 nM CB6 and 25 μM RVKR for 1 hour, then co-cultured with HEK293T cells expressing ACE2 for 16 hours. Blots are representative of two individual repeats; (**D**) Zsgreen fluorescent images of CB6-mediated neutralization of cell-cell fusion described in (C). Scale bars are indicative of 50 μm, images are representative of two independent experiments.
(TIF)

**S5 Fig. CB6-induced SARS-CoV-2 cell-cell transmission in Vero E6-TMPRSS2 cells.** (**A**) Immunofluorescent images showing morphology of 2 MOI SARS-CoV-2 infected Vero E6-TMPRSS2 cells, without or with 1 μg/mL Anti-ACE2 blocking antibody, treated without or with 12.5 nM CB6 antibody 48 hours post infection (hpi). Anti-SARS-CoV-2 N and Anti-human IgG (H+L chains) were stained with Alexa fluor 488 and 555 respectively. Scale bars are indicative of 50 μm; (**B**) Immunoblots of SARS-CoV-2 full-length spike, S2, S2' and N, collected from Vero E6-TMPRSS2 lysates described in (A), blots are representative of two individual repeats.
(TIF)

## Acknowledgments

We thank Qiuhong Guo, Prof. Gary Wong and Prof. Jin Zhong for their experimental supports and key reagents used in this work. We also thank Dr. Shuai Xia and Prof. Shibo Jiang for providing the peptide fusion inhibitor EK1C4.

## Author Contributions

**Conceptualization:** Shi Yu, Guangxun Meng.

**Data curation:** Shi Yu, Xu Zheng, Guangxun Meng.

**Formal analysis:** Shi Yu, Xu Zheng, Yuhui Gao, Guangxun Meng.

**Funding acquisition:** Shi Yu, Dianfan Li, Dimitri Lavillette, Guangxun Meng.

**Investigation:** Shi Yu, Xu Zheng, Yanqiu Zhou, Yuhui Gao, Yunyi Li, Jiabin Mou, Xiaoxian Cui, Yuying Yang.

**Methodology:** Shi Yu, Xu Zheng, Dimitri Lavillette, Guangxun Meng.

**Project administration:** Shi Yu, Xu Zheng, Guangxun Meng.

**Resources:** Shi Yu, Xu Zheng, Yanqiu Zhou, Bingjie Zhou, Yapei Zhao, Tingting Li, Yunyi Li, Jiabin Mou, Xiaoxian Cui, Yuying Yang, Dianfan Li, Min Chen, Dimitri Lavillette, Guangxun Meng.

**Software:** Shi Yu.

**Supervision:** Dimitri Lavillette, Guangxun Meng.

**Validation:** Shi Yu, Xu Zheng.

**Visualization:** Shi Yu, Xu Zheng.

**Writing – original draft:** Shi Yu, Guangxun Meng.

**Writing – review & editing:** Shi Yu, Xu Zheng, Guangxun Meng.

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
