## [Decision Letter · Decision Letter 0]

25 Jul 2023

Dear Dr. Meng,

Thank you very much for submitting your manuscript "Antibody-mediated spike activation promotes cell-cell transmission of SARS-CoV-2" for consideration at PLOS Pathogens. As with all papers reviewed by the journal, your manuscript was reviewed by members of the editorial board and by several independent reviewers. In light of the reviews (below this email), we would like to invite the resubmission of a significantly-revised version that takes into account the reviewers' comments.

Three expert reviewers have carefully reviewed your submission. Altogether there are many thoughtful recommendations to improve the manuscript. Several new experiments are recommended and the results of these tests are necessary to elevate the submission up to journal standards.

We cannot make any decision about publication until we have seen the revised manuscript and your response to the reviewers' comments. Your revised manuscript is also likely to be sent to reviewers for further evaluation.

Sincerely,

Tom Gallagher

Guest Editor

PLOS Pathogens

Guangxiang Luo

Section Editor

PLOS Pathogens

Kasturi Haldar

Editor-in-Chief

PLOS Pathogens

orcid.org/0000-0001-5065-158X

Michael Malim

Editor-in-Chief

PLOS Pathogens

orcid.org/0000-0002-7699-2064

Three expert reviewers have carefully reviewed your submission. Altogether there are many thoughtful recommendations to improve the manuscript. Several new experiments are recommended and the results of these tests are necessary to elevate the submission up to journal standards.

Reviewer's Responses to Questions

**Part I - Summary**

Reviewer #1: In this manuscript, the authors aimed to elucidate the mechanism of antibody-enhanced cell-cell fusion by SARS-CoV-2 spike (S). They first observed that S RBM antibodies (CB6 and REGN10933) induces S-mediated cell-cell fusion significantly in 293T cells, which expressed very low levels of receptor hACE2. These ab-induced cell-cell fusions correlated with other fusion phenotypes including S1-shedding and S2’ cleavage. Interestingly, ab-induced cell-cell fusion is reduced with S1/S2 cleavage-deficient S proteins (R685A, siFURIN, RVKR), and enhanced when S1/S2 cleavage is more extensive (FURIN OE). In high hACE2 expression conditions (293T OE hACE2), CB6’s neutralization of SARS-2 pps was S1/S2 cleavage-independent (both WT and R685A were neutralized), but the neutralization of cell-cell fusion was S1/S2 cleavage-dependent (R685A and RVKR were neutralized, while WT was not). The latter point was strengthened by the fact that trypsin addition can overcome the effect of R685A, making its cell-cell fusion ab-resistant (WT-like). The cell-cell fusion of deltaRRAR S, with its complete loss of cleavability at S1/S2, remains sensitive to ab. Lastly, the authors used authentic SARS-CoV-2 virus to validate the differential effects of RBM ab (CB6) on virus particles and cell-cell fusion. They found that ab enhanced CPE, S2’ cleavage, and N protein abundance in infected cells in a dose-dependent manner. Conversely, the output virus was neutralized by ab in a dose-dependent manner. While some of the phenotypes are not completely novel, this study carefully dissected the different phenotypes in hACE2 low vs high and cell entry vs cell-cell fusion conditions to a degree that one can start to grasp the bigger picture, but some conclusions are out of place. Overall, the study had documented some interesting phenotypes, but the results can get a bit confusing at times.

Reviewer #2: In this study, Yu and colleagues investigate how RBM-targeting antibodies, particularly CB6, can activate the spike protein resulting in cell-cell fusion. They demonstrate that binding of class I antibodies can trigger syncytia formation in several cell lines including 293T and A549 expressing low levels of ACE2. They also show that class I antibodies trigger increased S1 shedding that is accompanied by the spike cleavage including at the S2’ site. They show that the binding of class I antibody CB6 to the spike protein causes a conformational change that is likely associated with post-fusion state of the spike. They conclude that the phenotypes of increased syncytia formation, increased shedding, increased cleavage at S2’, and the conformational change are reliant on furin cleavage at the S1/S2 boundary by using a spike with an R685A change or a furin cleavage site deletion, as well as knockdown of furin or by an inhibitor against furin.

While overall the study is well designed and the data is interesting, there are critical weaknesses that must be addressed. First, the role of ACE2 in CB6-mediated cell-cell fusion cannot be completely ruled out, as the cell lines used, including 293T and A549, still express a low level of ACE2. The author should have completely depleted ACE2 in these cells, or use cell lines that are known to express no ACE2. Second, throughout the study, the surface expression of spike is not examined. This is important, given that cell surface expression of spike critically determines cell-cell fusion and is also impacted by S1 shedding. Third, in multiple locations, the author tries to link cell-cell fusion results to cell-cell transmission; however, cell-cell transmission is distinct from cell-cell fusion, and carefully designed cell-cell transmission assay shall be performed to support their conclusions. The use of “virus-to-cell transmission” in the text is also unconventional and confusing. Fourth, while live SARS-CoV-2 virus is used resulting in CPE in Vero E6 cells, no definitive conclusion can be drawn from this experiment, because no clear cell-cell fusion is observed in the presence of CB3, not it is tested in ACE2-deficient cells or using furin-deficient SARS-CoV-2 viruses. The author claim that the CB6 did not prevent the formation of CPE induced by SARS-CoV-2 virus, but the opposite was shown in Fig. 6. Given SARS-CoV-1 does not contain a furin cleavage site, which can serve a perfect control; unfortunately, SARS-CoV-1 is not tested nor mentioned throughout including in the Discussion. The discussion appears open-ended and should be extended to explain the mechanisms of action regarding role of furin, which is normally present in trans-Golgi but not on the plasma membrane, as well as implications for COVID infection and pathogenesis.

Reviewer #3: Yu and colleagues investigated the impact of antibodies targeting the receptor binding motif (RBM) of SARS-CoV-2 spike protein on S protein-driven cell-cell fusion. In brief, they report that certain RBM-specific antibodies induce S protein-driven cell-cell fusion as well as shedding of the S1 subunit und cleavage at the S2’ site. Furthermore, evidence for a role of furin in antibody-induced cell-cell fusion is reported and it is demonstrated that cleavage of the S protein at the S1/S2 site protects against blockade of cell-cell fusion by an RBM specific antibody. Finally, evidence is provided that an RBM specific antibody induces cell-cell fusion in the context of SARS-CoV-2 infected cells. The concept proposed is interesting. However, there are several major points that remain to be addressed.

**Part II – Major Issues: Key Experiments Required for Acceptance**

Reviewer #1: 1. Some of the conclusion sentences are inappropriate and make reading through the document quite difficult and quite confusing: 1) Line 175 mentions viral infection, when infection was not assessed there. 2) Line 195 mentions RBD abs are related to “cleavage at two proteolytic sites”, when only S2’ cleavage was assessed at this point. S1/S2 cleavage was assessed later. 3) Line 284 concludes that abs bind to S and causes S to be internalized, and this is the mechanism that prevents cell-cell fusion. However, internalization was not assessed here, nor was the causation of cell-cell fusion tested. 4) Line 351 states that abs can activate R685A and deltaRRAR S proteins only when trypsin was added (Fig 5C). This is incorrect because trypsin had no effect on deltaRRAR profile, and CB6 DID NOT activate cell-cell fusion beyond what was already achieved by trypsin. 5) Line 359 states that S1 shedding is responsible for ab-induced cell-cell fusion. This causation relationship was not tested, so please rephrase.

2. Throughout the manuscript (lines 193, 215, 220, 234, 236), the authors stated CB6 “induced the spike cleavage of S1”. Since none of the data suggested the abs actually induced more cleavage at S1/S2, but instead causes S1-shedding, please rephrase the sentences for accuracy.

3. Throughout the paper, cell-cell fusion correlated well with S2’ cleavage (as shown with WB), except for Fig 2E, where R685A barely fused cells together but it has just as much S2’ cleavage as WT. Please explain this disconnection.

4. Please provide a graphical abstract-style figure at the end of the paper. The detail points of this paper can get very confusing. Also, the authors did not propose a mechanism that explains these complicated phenotypes in hACE2-low vs low cells, and in particle neutralization vs cell-cell fusion promotion. Both of these points can be clarified tremendously by adding a conclusion schematic explaining the findings and the proposed mechanism.

Reviewer #2: 1. Throughout the manuscript, no quantifications of western blotting and microscopic imaging are performed. While S1 shedding data is shown, the S cleavage with ratios of both S1/S and S2/S in cell lysates should be examined and shown in parallel.

2. Fig. 2C, no cell lysate data is presented, which is critical. Based on Fig. 2A in 293T cells, CB6 didn’t induce the spike cleavage at S1/S2 site; the author should check and ensure all statements are correct throughout the manuscript.

3. Fig. 6: this is one of the most critical experiments, but unfortunately it was not carefully designed – why was Vero E6 cell chosen? And would how the results from Vero E6 cells reflect that from cells expressing low ACE2? Was the virus produced from Vero E6 sequenced to confirm that there is no deletion in the S1/S2 cleavage and other sites? And how S1 shedding correlated with CPE or cell-cell fusion?

Reviewer #3: The manuscript is on several occasions difficult to read and revision for grammar and style issues is needed.

It is stated on several occasions that the CB6 antibody induces S protein cleavage at the S1/S2 site. However, this statement is not supported by data. Thus, it is demonstrated that CB6 induces shedding of the S1 subunit and S protein cleavage at the S2’ site but no evidence for CB6 induced cleavage at S1/S2 is reported.

The authors state on several occasions the WT 293T cells were used as target cells in the cell-cell fusion assay to demonstrate ACE2-independent cell-cell fusion. 293T cells express endogenous ACE2, although some batches at low levels, and are not suitable to demonstrate ACE2 independent cell-cell fusion. For this, experiments with ACE2 KO 293T cells and anti-ACE2 antibodies that block S protein binding to ACE2 are essential.

It is important to show whether the effects of the antibodies CB6 and REGN10987 in figure 1B were concentration dependent. Furthermore, appropriate controls must be shown: Signals obtained with effector cells alone, target cells alone and effector cells without spike mixed with target cells.

Figure 3C-E: It is essential to compare blockade (and overexpression, respectively) of furin in effector cells and target cells with the expectation being that furin blockade in target cells blocks CB6 induced cell-cell fusion. However, from the results section and the figure legends it is unclear what has been done. Finally, it is essential to not only quantify cell-cell fusion but to also investigate cleavage at the S2’ site under these conditions.

Figure 4D: It is well established that R685R markedly reduces S protein-driven cell-cell fusion. Were conditions chosen that allowed for comparable cell-cell fusion driven by WT and R685A S proteins in the absence of antibody? If not, is the inhibition of R685 S protein-driven cell-cell fusion by CB6 simply due to the overall low level of fusion activity?

**Part III – Minor Issues: Editorial and Data Presentation Modifications**

Reviewer #1: 1. The word “auto-process” was used to describe S1/S2 cleavage. Since S protein does not cleave itself (aka “auto), I suggest omitting the word “auto”. This appeared a few times.

2. HEK293T is not hACE2 KO, but instead has a very low level of hACE2. Please acknowledge this in the manuscript.

3. The paper uses a collection of antibodies (abs), all used at the same dose. What are the affinity differences of these abs, and can these differences potential explain some of the differential effects documented? Please acknowledge.

4. The IFA in Fig 1C is too difficult to independently interpret. Maybe this suggests the ab enhancement of cell-cell fusion is modest in the presence of adequate levels of hACE2. Maybe consider relocating this panel elsewhere.

5. Fig 1E and 2D. What are the mCherry+ cells (look like singular cells) in the absence of ab? Please acknowledge.

6. Line 194 mentions Fig 2C to contain lysate data. I do not see lysate data.

7. Line 298 is stated incorrectly. CB6 did not inhibit S1-shedding, RVKR did.

8. Line 310 concludes that “trypsin alone triggers minimal S1 shedding”, but “minimal” relative to what? Fig 5B does not have a positive control for S1-shedding, hence this conclusion is inappropriate. Should consider rewording or eliminating this statement.

9. Line 389 states that this paper shows ab-induced S activation is a “significant biological” feature. This paper does not demonstrate “biological”. Please rephrase.

Reviewer #2: 4. A number of statements are not supported by experimental evidence; these include: ACE2-deficient HEK293T cells; role of K417N and Q493R in BA1/BA4 binding to CB6; antibody-mediated spike activation “exclusively” promoting the cell-cell transmission of SARS-CoV-2, etc.

5. There is no detailed explanations or discussion regarding S2’ cleavage, which could be mediated by TMPRSS2 and/or Cathepins.

6. Fig. 3C-D: it’s confusing here that the author did not mention about CB6 but included ACE2 in the legend. Are experiments performed in ACE2 overexpressing cells as done in Fig. 4? The conclusion could be different using ACE2 overexpression. Western blotting of cell lysate should be also shown.

7. Fig. S3: why are WT �19 and Delta �19, rather than their native spikes used? An explanation should be given.

8. Where is EK1C4 peptide used in this study? It is mentioned in the paper.

9. Some incorrect or overstatements should be modified or corrected:

a. “Others have also reported that spike-mediated cell-cell fusion is resistant to neutralization antibodies, convalescent and vaccinated sera at doses higher than IC90 concentration of virion particles [31, 32, 47, 49]” – this is not true for cell-cell fusion mediated by spike.

b. “furin cleavage site strongly promotes cell-cell transmission of the virus” – there is no perfect correlation between them, proper references should be provided.

c. “it is well-known that cell-cell fusion between lymphocytes driven by the HIV-1 envelop is insensitive to neutralization by antibodies or patient anti-sera” – this is not always true.

d. “The presence of CB6 antibody had no effect on intracellular viral replication and did not prevent the formation of CPE induced by SARS-CoV-2 virus, yet the antibody facilitated the spike cleavage and cell-cell fusion” – this is an overstatement with no convincing data.

Reviewer #3: “The latest human codon optimized Omicron was purchased by Genescripts” Please specify.

Please provide source and references for the plasmids used to quantify cell-cell fusion.

PLOS authors have the option to publish the peer review history of their article (what does this mean?). If published, this will include your full peer review and any attached files.

Reviewer #1: No

Reviewer #2: No

Reviewer #3: No
---

## [Decision Letter · Decision Letter 1]

30 Oct 2023

Dear Dr. Meng,

We are pleased to inform you that your manuscript 'Antibody-mediated spike activation promotes cell-cell transmission of SARS-CoV-2' has been provisionally accepted for publication in PLOS Pathogens.

Best regards,

Tom Gallagher

Guest Editor

PLOS Pathogens

Guangxiang Luo

Section Editor

PLOS Pathogens

Kasturi Haldar

Editor-in-Chief

PLOS Pathogens

orcid.org/0000-0001-5065-158X

Michael Malim

Editor-in-Chief

PLOS Pathogens

orcid.org/0000-0002-7699-2064

Reviewer Comments (if any, and for reference):

Reviewer's Responses to Questions

**Part I - Summary**

Reviewer #1: The authors have addressed enough of my question. No further requests.

Reviewer #3: The authors have vigorously responded to the critique raised by this reviewer. The revised manuscript has been significantly improved.

**Part II – Major Issues: Key Experiments Required for Acceptance**

Reviewer #1: The authors have addressed enough of my question. No further requests.

Reviewer #3: None

**Part III – Minor Issues: Editorial and Data Presentation Modifications**

Reviewer #1: The authors have addressed enough of my questions. No further requests.

Reviewer #3: None

PLOS authors have the option to publish the peer review history of their article (what does this mean?). If published, this will include your full peer review and any attached files.

Reviewer #1: No

Reviewer #3: No

---

## [Editor Report · Acceptance letter]

6 Nov 2023

Dear Professor Meng,

We are delighted to inform you that your manuscript, "Antibody-mediated spike activation promotes cell-cell transmission of SARS-CoV-2," has been formally accepted for publication in PLOS Pathogens.

Best regards,

Kasturi Haldar

Editor-in-Chief

PLOS Pathogens

orcid.org/0000-0001-5065-158X

Michael Malim

Editor-in-Chief

PLOS Pathogens

orcid.org/0000-0002-7699-2064